# Dynamic restructuring of nickel sulfides for electrocatalytic hydrogen evolution reaction

Xingyu Ding [1,2,8], Da Liu [2,8], Pengju Zhao[1], Xing Chen[1], Hongxia Wang[1], Freddy E. Oropeza [3] ✉, Giulio Gorni[4,5], Mariam Barawi[3], Miguel García-Tecedor [3], Víctor A. de la Peña O'Shea [3], Jan P. Hofmann [6], Jianfeng Li [1], Jongkyoung Kim[7], Seungho Cho[7], Renbing Wu [2] ✉ & Kelvin H. L. Zhang [1] ✉

Transition metal chalcogenides have been identified as low-cost and efficient electrocatalysts to promote the hydrogen evolution reaction in alkaline media. However, the identification of active sites and the underlying catalytic mechanism remain elusive. In this work, we employ operando X-ray absorption spectroscopy and near-ambient pressure X-ray photoelectron spectroscopy to elucidate that NiS undergoes an in-situ phase transition to an intimately mixed phase of $Ni_3S_2$ and NiO, generating highly active synergistic dual sites at the $Ni_3S_2$/NiO interface. The interfacial Ni is the active site for water dissociation and OH* adsorption while the interfacial S acts as the active site for H* adsorption and $H_2$ evolution. Accordingly, the in-situ formation of $Ni_3S_2$/NiO interfaces enables NiS electrocatalysts to achieve an overpotential of only 95 ± 8 mV at a current density of 10 mA cm$^{-2}$. Our work highlighted that the chemistry of transition metal chalcogenides is highly dynamic, and a careful control of the working conditions may lead to the in-situ formation of catalytic species that boost their catalytic performance.

Electrolysis of water powered by renewable electricity to produce green hydrogen is widely considered as a promising pathway to a global clean and sustainable energy future[1,2]. The key to enable this technology is the development of low-cost and highly efficient electrocatalysts to accelerate the water splitting reactions, i.e., oxygen evolution reaction (OER) and hydrogen evolution reaction (HER)[3]. In particular, the design of high-performance electrocatalysts for HER in alkaline conditions has received intensive attention, because the alkaline water electrolysis is the most commonly used route in the industry, and the alkaline HER is also a key step in the chlor-alkali process[4,5]. However, the alkaline HER still suffers from sluggish reaction kinetics compared to that in acidic

solution, due to the additional water dissociation step (Volmer step: $H_2O + M^* + e^- \rightarrow M^*H + OH^-$) in order to supply the absorbed H* intermediate for subsequent $H_2$ generation[6]. The water dissociation has a higher activation barrier and is considered to be the rate determining step that limits the overall reaction[6,7]. Even the state-of-the-art Pt-based catalysts show about two orders of magnitude lower HER activity in alkaline than in acidic media[3,8,9]. Significant progress has been made to improve the HER activity of Pt and other noble metals in alkaline environments by constructing dual active site heterostructures, e.g., Pt-Ni(OH)$_2$, in which the Ni(OH)$_2$ functions as a promotor to accelerate the water dissociation[10]. Improving the activity of noble metal-based

[1]State Key Laboratory of Physical Chemistry of Solid Surfaces, College of Chemistry and Chemical Engineering, Xiamen University, Xiamen 361005, China. [2]Department of Materials Science, Fudan University, Shanghai 200433, China. [3]Photoactivated Processes Unit, IMDEA Energy Institute, Parque Tecnológico de Móstoles, Avda. Ramón de la Sagra 3, 28935 Móstoles, Madrid, Spain. [4]Laser Processing Group, Institute of Optics (CSIC), C/Serrano 121, 28006 Madrid, Spain. [5]CELLS-ALBASynchrotron, Carrer de la Llum 2-26, 08290 Cerdanyola del Vallès, Spain. [6]Surface Science Laboratory, Department of Materials and Earth Sciences, Technical University of Darmstadt, Otto-Berndt-Strasse 3, 64287 Darmstadt, Germany. [7]Department of Materials Science and Engineering, Ulsan National Institute of Science and Technology (UNIST), Ulsan 44919, Republic of Korea. [8]These authors contributed equally: Xingyu Ding, Da Liu. ✉e-mail: freddy.oropeza@imdea.org; rbwu@fudan.edu.cn; kelvinzhang@xmu.edu.cn

catalysts with a single active site has been challenging mainly due to the scaling relationships in the formation energy of intermediates, which affects not only the HER but also the OER and the oxygen reduction reaction (ORR)[11,12]. Dual active site catalysts provide alternative reaction mechanisms that may break such scaling relationships, offering more chance for achieving better catalytic performance, compared with single active site catalysts[12,13].

Nevertheless, the scarcity and high cost of Pt group materials restrict their large-scale application[14]. In this regard, earth-abundant transition metal (TM)-based chalcogenides, phosphides, nitrides, and carbides have been recently emerging as cost-effective alkaline HER electrocatalysts[15–21]. Many TM-based catalysts exhibit comparable alkaline HER activity as Pt group materials[22]. In addition, doping, defect, and strain engineering have been further explored to modulate their electronic structure to enhance the activity and stability[23–27]. Nickel sulfides such as NiS, $NiS_2$, and $Ni_3S_2$ have drawn significant attention because of their high catalytic activity, as well as easy and scalable methodologies for the preparation[28–30]. Despite this rapid development of alkaline HER catalysts, the underlying reaction mechanism and the active sites for the alkaline HER, which are different from the well-documented acidic HER, are still under considerable debate[6,7,31]. Furthermore, it has been demonstrated that TM-based compounds catalysts for the OER undergo structural or compositional reconstruction, or even transform into a new phase under OER working condition[32–36]. On the other hand, the structural and chemical conversion of catalysts under HER conditions has been less explored, and it has been even assumed HER catalysts are stable during the HER process[37]. However, recent works indicated that

some transition metal chalcogenides may indeed undergo a phase transition under alkaline HER working condition[38–40]. Therefore, real-time monitoring of the catalytic processes and identifying the real active sites under HER operation conditions are of vital importance to uncover the catalytic mechanism.

In this study, we present an in-depth elucidation of the phase conversion process based on operando X-ray absorption spectroscopy (XAS) and operando Raman spectroscopy under HER working conditions. We found that NiS electrodes undergo a significant HER performance enhancement due to an in-situ transformation into a mixed phase of $Ni_3S_2$ and NiO, which is identified to be the actual active catalyst phase for alkaline HER. The ex-situ spectroscopy and theoretical investigations further reveal that the interfacial Ni (O-**Ni**-S) of $Ni_3S_2$/NiO provides the active sites to accelerate water dissociation while the interfacial S (O-Ni-**S**) facilitates the adsorption of hydrogen intermediates and its subsequent association to form molecular hydrogen. As a result, the in-situ formed $Ni_3S_2$/NiO exhibits a superior HER activity, requiring a low overpotential of $95 \pm 8$ mV to reach a current density of 10 mA cm$^{-2}$ in our performance tests.

## Results and discussion
### Phase transition of NiS to $Ni_3S_2$/NiO induces enhanced HER activity
The NiS samples were synthesized on carbon paper using one-step hydrothermal method (Supplementary Fig. 1a). Details of the synthesis and characterization are provided in the Methods section. X-ray diffraction (XRD) pattern shown in Fig. 1a and X-ray photoelectron

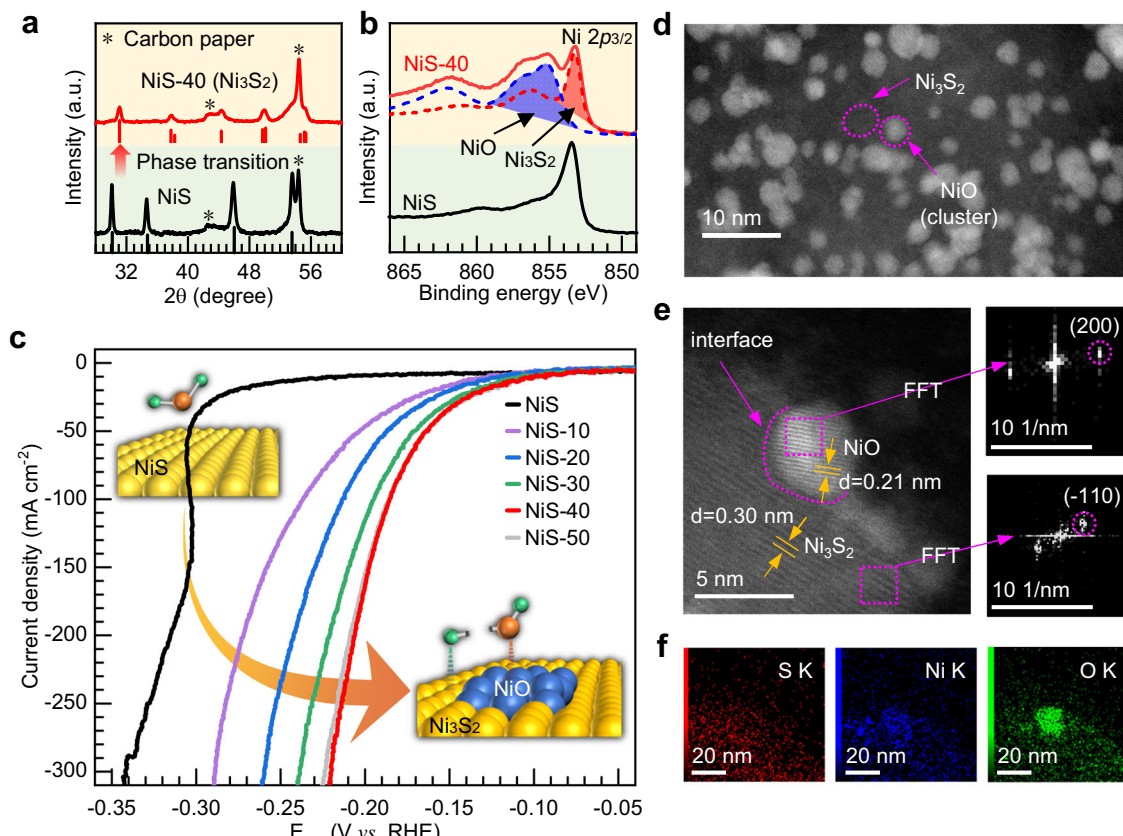

**Fig. 1 | HER performance and structural characterizations of NiS catalyst before and after measurement. a** XRD patterns of the as-synthesized NiS and NiS after HER measurement (marked as NiS-40). **b** Ni 2*p* XPS spectra for NiS and NiS after HER measurement (marked as NiS-40), along with NiO and $Ni_3S_2$ as reference. **c** *iR*-corrected Linear sweep voltammetry (LSV) polarization curves of as-synthesized NiS, NiS-10, NiS-20, NiS-30, NiS-40, and NiS-50 samples measured in

1 M KOH solution. NiS-10, NiS-20, NiS-30, NiS-40, and NiS-50 are NiS samples after 10, 20, 30, 40, and 50 minute chronoamperometry measurements at −1.00 V vs. RHE, respectively. **d** The large-area TEM image, **e** high-resolution TEM image (the right panel shows the FFT images of both regions) and **f** the corresponding elemental mapping of the NiS-40 sample after conducting 25-h chronopotentiometric measurement at a current density of 200 mA cm$^{-2}$.

spectroscopy (XPS) of Ni 2$p$ core level in Fig. 1b suggests that the as-synthesized samples are characteristic of the hexagonal α-phase of NiS[28]. Scanning electron microscopy (SEM) and transmission electron microscopy (TEM) images in Supplementary Fig. 1 show that the samples consist of NiS particles with a size of ~200 nm uniformly coating on the carbon paper substrate.

The HER activity was evaluated in 1 M KOH using a three-electrode system (details in the Methods section). Cyclic voltammetry (CV) measurements were carried out in order to adjust the conditions for the evaluation of the HER catalytic activity (Supplementary Fig. 2a, b). In the first cycle of as-synthesized samples, there is a sharp onset of the current density for the HER at around −0.30 V vs. RHE, which continues to increase steadily as the sample is further polarized to negative potentials. After that sharp onset of the current density, the CV profile is reproducible in the second cycle, with an upper limit of the potential at 0.05 V vs. RHE. However, as the upper limit of the potential is extended to 0.40 V vs. RHE in the 3rd and 4th cycles, there is an oxidation wave with a peak at +0.27 V vs. RHE and a 15% increase of the current density when the sample is polarized back to negative potentials. Remarkably, the current density keeps on increasing as the sample is subject to chronoamperometry at −1.00 V vs. RHE (Supplementary Fig. 2c). Linear sweep voltammetry (LSV) of the sample after the chronoamperometry measurement shows that the current density increases 25% compared with the pristine sample, and there is also a clear shift of the onset to lower overpotential (Supplementary Fig. 2d). These electrochemical studies clearly demonstrate that in-situ modifications of NiS electrodes lead to a significantly enhanced HER catalytic activity.

On these bases, we carried out a more detailed study of the chronoamperometric conditions of NiS electrodes on their HER catalytic activity. Figure 1c shows the LSV curves of the as-synthesized NiS, as well as NiS samples after 10, 20, 30, 40, and 50 minute chronoamperometry measurements at −1.00 V vs. RHE, labeled as NiS-10, NiS-20, NiS-30, NiS-40, and NiS-50, respectively. In order to avoid deviations caused by ohmic drops, the potential was adjusted by an $iR$ factor (Supplementary Fig. 3a, b). The as-synthesized NiS initially exhibits a low HER activity, but the HER activity gradually increases with the chronoamperometry measurement time until it is conducted for 40 minutes. A similar trend of HER activity is also observed when the current density is normalized by electrochemical active surface area (ECSA) (Supplementary Figs. 4 and 5). As shown in Fig. 1c, the NiS-40 sample exhibits a high HER catalytic activity, requiring an overpotential of 95 ± 8 mV to reach a current density of 10 mA cm$^{-2}$ (acquisition of error bar can be seen in Supplementary Fig. 3c.). Further electrochemical characterizations showed that the Tafel slope and charge transfer resistance (Supplementary Fig. 6a, b) decrease in the order of NiS > NiS-10 > NiS-20 > NiS-30 > NiS-50 > NiS-40, indicating that the performance enhancement after the chronoamperometric treatments results from an intrinsic faster HER kinetics. Compared with previously reported HER catalysts, electrochemically treated NiS electrocatalysts exhibit competitive HER activity in terms of overpotentials at 10 mA cm$^{-2}$ and Tafel slope (Supplementary Fig. 7 and Supplementary Table 1). Additionally, the catalytic activity of the NiS-40 sample is very stable, showing no overpotential increase over 25 h of continuous operation at a current density of 200 mA cm$^{-2}$ in 1 M KOH (Supplementary Fig. 6c).

A series of ex-situ characterizations of the highly active samples after performance tests were conducted. The as-synthesized NiS is converted into a mixed phase of Ni$_3$S$_2$ and NiO after HER measurement. Figure 1a shows an XRD pattern of the NiS-40 sample, from which a pattern characteristic of Ni$_3$S$_2$ can be identified, suggesting a full-phase conversion of NiS to crystalline Ni$_3$S$_2$[41,42]. The XPS in the Ni 2$p_{3/2}$ region for the NiS-40

sample exhibits a pronounced spectral feature possibly arising from NiO (Fig. 1b). Additionally, the Ni K-edge XAS of sample NiS-40 (Supplementary Fig. 8) can be fitted by a linear combination of spectra of Ni$_3$S$_2$ and NiO, which further confirms the formation of Ni$_3$S$_2$ and NiO. We further characterized the NiS-40 sample after 25-h chronopotentiometric measurement at a current density of 200 mA cm$^{-2}$, the XRD, XPS and XAS confirm the HER-post catalysts are present in the form of Ni$_3$S$_2$ and NiO (Supplementary Fig. 9). Spherical Aberration Corrected Transmission Electron Microscope (SAC-TEM) was used to further examine the microstructure of the HER-post catalysts. As shown in Fig. 1d, the large-area TEM image reveals numerous small NiO clusters anchored onto the large Ni$_3$S$_2$. The HR-TEM image exhibits a distinct interface that partitions the image into two regions (Fig. 1e). By analyzing the FFT images corresponding to these two regions, the cluster region has an interplanar spacing of 0.21 nm corresponding well to the (200) lattice plane of NiO, while the other region has an interplanar spacing of 0.30 nm which matches well with (−110) lattice plane of Ni$_3$S$_2$. More importantly, the elemental mapping shows the preferential enrichment of oxygen elements on the small cluster (Fig. 1f), which strongly reveals the small cluster is assigned to NiO. It should be noted that the formed NiO may exist in the form of a very small cluster, which is hard to determine by XRD patterns. To investigate the influence of the ratio of Ni$_3$S$_2$ to NiO on the catalytic activity, NiS samples received by 1, 5, 10, and 50 LSVs in the range from 0.00 V to −1.00 V vs. RHE were prepared. The Ni K-edge XAS of these samples, shown in Supplementary Figs. 10a−d, consists of a linear combination of spectra of Ni$_3$S$_2$ and NiO, which indicates that a full transformation of NiS occurs after a single LSV. However, the NiO to Ni$_3$S$_2$ ratio increases upon increasing the number of LSVs. We extracted the NiO to Ni$_3$S$_2$ ratio based on the XAS spectra (Supplementary Figs. 10a−d and Supplementary Table 2) and found that the HER activity increases with the amount of NiO (Supplementary Fig. 10e), which clearly indicates the beneficial effect of the Ni oxide species on the HER process.

## Phase transition dynamics revealed by operando XAS and operando Raman spectroscopy

An operando XAS spectroscopy was employed to gain detailed insights into the NiS→Ni$_3$S$_2$/NiO transition process, and details of the experimental method are provided in the Methods section. Figure 2a shows a three-dimensional (3D) plot of the Ni K-edge XAS of the NiS electrode as a function of the applied potentials during cyclic CV measurement (Fig. 2b). Note that two important changes in the spectral profile occur at applied voltages of −0.27 V and +0.27 V vs. RHE, dividing the 3D plot into three regions as I (applied voltage from +1.00 V to −0.27 V), II (−0.27 V to −1.50 V and back to +0.27 V) and III (from +0.27 V to 1.00 V). Selected Ni K-edge XAS spectra at specific applied voltages within regions I, II and III are shown in Fig. 2c. In region I, with applied voltage from +1.00 V to −0.27 V, the spectrum of the sample remains as that of the NiS phase. The transition from region I to region II at −0.27 V can be identified as the phase transition NiS→Ni$_3$S$_2$, based on spectral fitting and comparisons with reference samples of NiS and Ni$_3$S$_2$ (region I and II in Fig. 2c). The associated reduction of the formal oxidation state of Ni$^{2+}$ in NiS to Ni$^{1.33+}$ in Ni$_3$S$_2$ can be observed as a cathodic wave peak in the CV profile just at the onset of the HER current shown in Fig. 2b (see the inset), clearly showing that the HER catalytic activity is associated with the formation of Ni$_3$S$_2$. After the phase transition, the Ni$_3$S$_2$ phase is stable under HER conditions. However, with a more negative voltage applied (region II in Fig. 2a), a continuous decrease of the spectral intensity of the white line of the Ni K-edge XAS spectra suggests a further reduction of the oxidation state of Ni. Consistently, the fitting of Fourier Transform moduli of spectra in region II revealed a decrease of the average coordination number of Ni coordinated by S (Fig. 2d, Supplementary Figs. 11 and 12, and Supplementary Tables 3 and 4, and

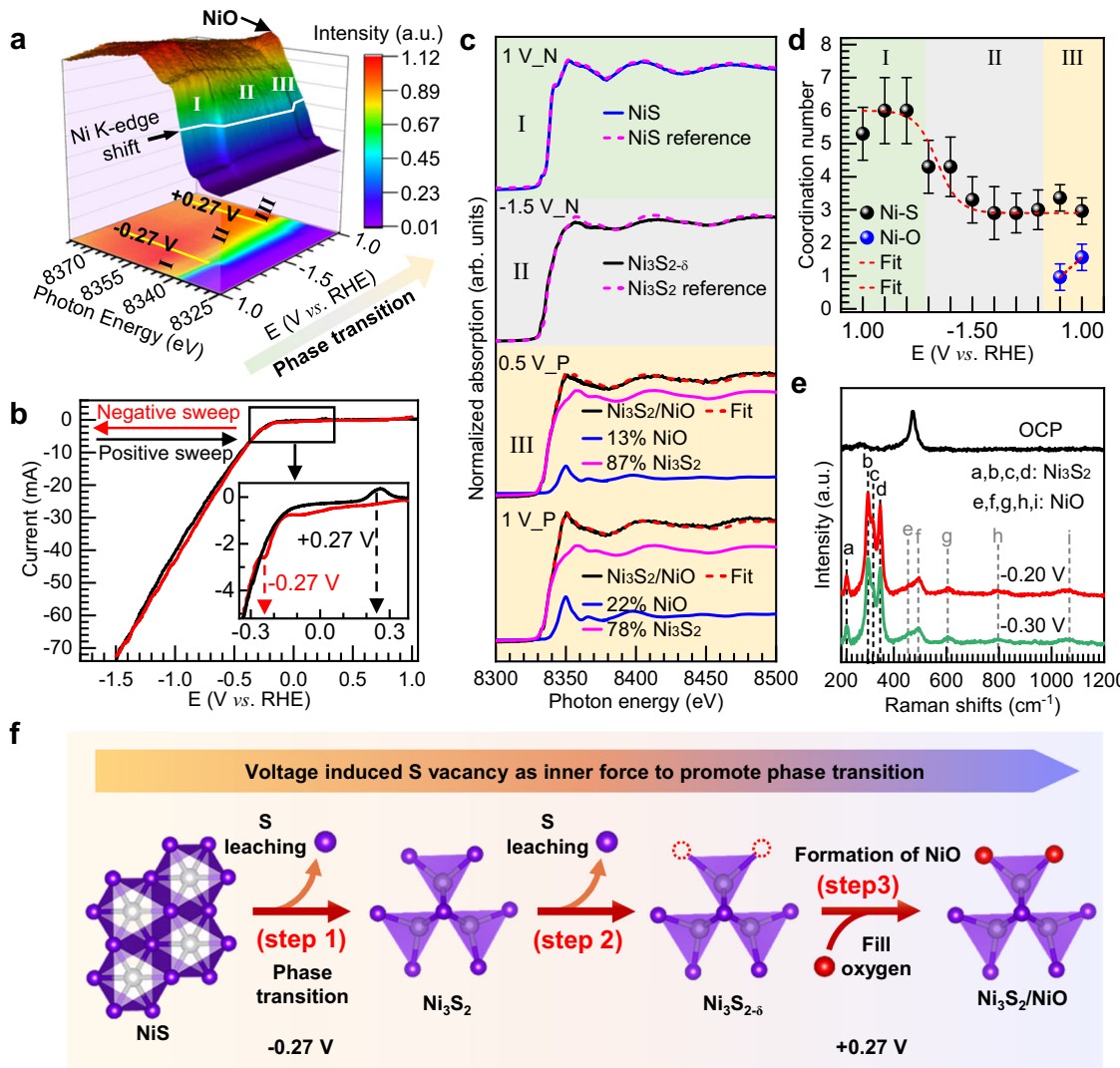

**Fig. 2 | Operando characterizations and the mechanism of phase transition.** **a** Operando Ni K-edge XAS for NiS catalyst as a function of applied voltages. Top panel shows a 3-dimensional view of Ni K-edge XAS spectra; bottom shows a 2-dimensional projection from a 3-dimensional view. **b** The corresponding CV profile used for the operando XAS measurement. The inset shows the magnified region marked in the rectangular box. **c** The Ni K-edge XAS spectra at specific applied voltages of +1.00 V (1 V_N means +1.00 V in negative sweep of CV), −1.50 V (−1.50 V_N means −1.50 V in negative sweep of CV), and +0.50 V, +1.00 V (+0.50 V_P and +1.00 V_P mean +0.50 V and +1.00 V in positive sweep of CV) in region I, II, and III respectively, extracted from Fig. 2a, and their linear combination of $Ni_3S_2$ and NiO. **d** The coordination number of Ni as a function of applied voltages. **e** Operando Raman spectroscopy of NiS at different applied voltages. **f** Schematic illustration for phase transition of NiS catalyst during HER measurement.

the simulations procedure in Supplementary Note 1), suggesting the formation of S vacancies, i.e., $Ni_3S_{2−δ}$. As the sample is polarized from negative back to positive potentials, the transition from II to III at +0.27 V can be associated with an oxidation process observed in the CV profile shown in Fig. 2b, consistent with a constant increase of the white line of the spectrum as the sample is further polarized (region III in Fig. 2a). The spectrum can be fitted with a linear combination of $Ni_3S_2$ and an increasing contribution of NiO (region III in Fig. 2c), and details of the fitting are discussed in Supplementary Notes 2–5. Based on this analysis, we estimate that the NiO contribution at +0.50 V is about 13%, which further increases to 22% when the sample potential reaches +1.00 V (Fig. 2c and Supplementary Table 5). Thus, the coordination of Ni by O occurs gradually in region III (Fig. 2d), probably caused by the filling of the sulfur vacancies in $Ni_3S_{2−δ}$ with water or OH⁻ ions, leading to an intimately mixed phase of $Ni_3S_2$ and NiO, which is further verified by the Wavelet transform (WT) analysis of extended X-ray absorption fine structure (EXAFS) and Fourier-transformed $k^3$-weighted EXAFS at applied voltage of +1.00 V in region III

(Supplementary Fig. 13). The corresponding results of fitting at +1.00 V were shown in Supplementary Table 4, and the Ni-O show a bond length of ~2.00 Å. Highly crystalline NiO has a characteristic Ni-O distances around 2.09 Å; however, the short-range order in NiO reveals that the Ni-O distances for amorphous and/or nanocrystalline NiO may decrease to 2.00–2.06 Å[43,44]. This result further confirms the formed NiO is in the form of a very small cluster.

The formation of mixed-phase of $Ni_3S_2$ and NiO was further verified by an operando Raman spectroscopy study (the details of the experiment in the Methods section). The top Raman spectrum in Fig. 2e can be assigned to pristine NiS, with characteristic bands appearing at Raman shifts 281 cm⁻¹ and 473 cm⁻¹. As the sample was polarized to negative potentials, the characteristic peaks assigned to NiS disappear, while the characteristic peaks assigned to $Ni_3S_2$ appear at 221, 303, 322, and 349 cm⁻¹ (marked as a, b, c, and d, respectively), indicating the complete phase transition from NiS to $Ni_3S_2$[45]. In addition, the newly emerged peaks e, f, g, h, and i located at 453, 493, 600, 800, and 1060 cm⁻¹ can be assigned to Ni-O[46–49], suggesting the

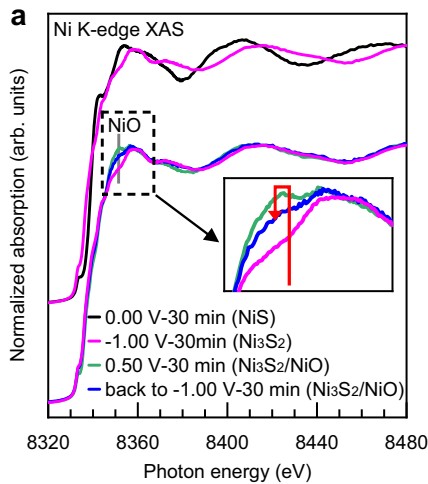

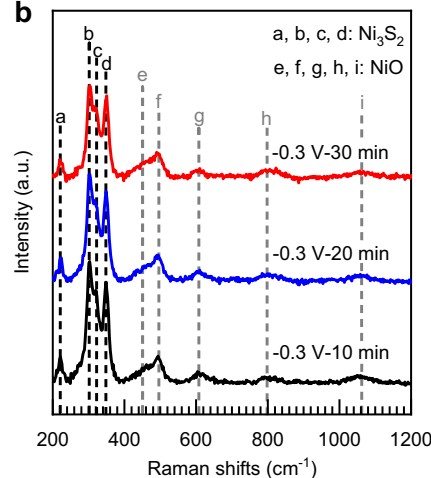

**Fig. 3 | The long-time existence of reconstructed Ni₃S₂/NiO proven by operando characterizations. a** Operando XAS at Ni K-edge of a NiS electrode in chronoamperometry for 30 minutes at 4 consecutively applied potentials: 0.00 V, −1.00 V, 0.50 V, and back in −1.00 V. **b** Operando Raman spectra of NiS electrode in chronoamperometry for 30 minutes at a constant applied potential of −0.30 V. (All potentials vs. RHE).

formation of NiO under HER conditions. Raman spectra features associated with NiS, Ni₃S₂, and NiO are very well resolved so that even small trace concentrations of NiO species can be detected.

Based on the analysis of the operando XAS and operando Raman spectroscopy data, we propose a three-step mechanism for the phase transition of NiS→Ni₃S₂/NiO under alkaline HER conditions, as schematically shown in Fig. 2f[50]. The electrochemical reduction of NiS to Ni₃S₂ at −0.27 V vs. RHE can be driven by the thermodynamic instability of NiS under HER conditions (step 1). As more negative potentials are set to reach higher HER current densities, S leaching leads to the formation of S vacancies in Ni₃S₂₋δ (step 2). As the sample is polarized to more positive potentials, substantial oxidation occurs at +0.27 V vs. RHE, and S vacancies are occupied by the oxygen species from the media (water/OH⁻ ions), leading to the formation of an intimately mixed phase of Ni₃S₂/NiO (step 3).

The stability of NiS, Ni₃S₂, and NiO under different conditions relevant to the HER catalytic performance was further investigated by operando XAS spectroscopy. Figure 3a shows operando XAS at Ni K-edge of the NiS electrode in chronoamperometry for 30 minutes at consecutively applied potentials: 0.00 V, −1.00 V, 0.50 V, and back to −1.00 V (all potentials are vs. RHE). As expected, the sample retains its pristine NiS structure at 0.00 V and converts to Ni₃S₂ when polarized to −1.00 V (Fig. 3a), due to the electrochemical reduction at −0.27 V as discussed above. Then, as the sample is polarized to +0.50 V, the white line of 8350 eV assigned to Ni-O increases noticeably (the inset in Fig. 3a) and the spectrum can be fitted with a linear combination of 90% of Ni₃S₂ features and 10% of NiO spectral features (Supplementary Fig. 14a and Supplementary Table 6), due to the bulk oxidation that occurs at +0.27 V as previously discussed. When the sample is brought back to HER conditions at −1.00 V, the content of NiO seems to decrease but still exist (the inset in Fig. 3a) as the spectrum can be fitted with a linear combination of 94% of Ni₃S₂ spectral features and 6% of NiO spectral features after 30 minutes under the reaction (Supplementary Fig. 14b and Supplementary Table 6). In addition, Fig. 3b shows operando Raman spectra of a NiS electrode in chronoamperometry at −0.30 V vs. RHE for 30 minutes, further confirming the existence of NiO under the HER process. In order to demonstrate the long-term stability of the Ni₃S₂/NiO mixed phase, we extended the operando Raman experiment at an applied potential of −0.30 V vs. RHE to 25 h. The results demonstrate Ni₃S₂ and NiO remain stable under HER conditions over an extended period of time (Supplementary Fig. 15).

## Insight into the HER mechanism

The HER in alkaline media mainly involves the step for water dissociation to form H* and OH*, i.e., Volmer step, followed by the coupling of H* to form H₂ in the Heyrovsky and Tafel steps. The Volmer step has been shown to be the rate-determining step for alkaline HER[6,7]. Therefore, to identify the active sites formed after the electrochemical treatment of the samples, we carried out a study for the dissociative adsorption of water on NiS and NiS-40 (Ni₃S₂/NiO) catalysts by means of near-ambient-pressure (NAP)-XPS (experimental details in the Methods section). NAP-XPS is a commonly used spectroscopic technique to study the dissociative adsorption of water on catalysts in a humid environment[51,52]. Figure 4b shows the XPS spectra of NiS and NiS-40 (Ni₃S₂/NiO) in the O 1s region in the presence of 0.5 mbar water vapor. Oxygen related species including lattice oxygen in NiO (lattice O), adsorbed OH*, adsorbed H₂O* and vapor H₂O (H₂O_vap) can be identified, and their relative amounts can be quantified by fitting the peak area (Supplementary Fig. 16 and Supplementary Table 7). In particular, the adsorbed OH* originates from the dissociative H₂O adsorption, and therefore can be used as a measure of the degree of water dissociation on the catalyst surface, i.e., the higher the value, the better the ability to dissociate water. As shown in Fig. 4c, the ratio ($R_{dis}$) of OH* to (H₂O* + OH*) in NiS-40 (Ni₃S₂/NiO) is much higher than that in NiS in the presence of different water vapor pressure, suggesting a stronger ability of dissociative adsorption of water on the NiS-40 (Ni₃S₂/NiO). In addition, the surface chemistry of the catalyst was further investigated by monitoring changes in the Ni 2p and S 2p core levels upon water exposure. For the NiS-40 (Ni₃S₂/NiO) sample, water exposure induces a pronounced change in the Ni $2p_{3/2}$ and S 2p. As shown in Fig. 4d, compared with the Ni $2p_{3/2}$ spectrum in vacuum, the spectral intensity associated with Ni-O increases significantly upon water exposure, which may result from the preferential adsorption of OH* at Ni sites. On the other hand, for S 2p shown in Fig. 4e, after water exposure, a shoulder feature appears at the lower binding energy side, which may result from the reduction of sulfur ions when H* adsorbs at S sites in Ni₃S₂. In contrast, for NiS (Supplementary Fig. 17), there is no obvious change in Ni 2p and S 2p after water exposure, which suggests that there is much less OH* and H* intermediates generated on NiS surfaces, in accordance with the weak water dissociation ability of NiS observed for this sample.

Considering the involvement of Ni and S sites in the water dissociation process, the influence of each site on the alkaline HER was investigated by a systematic poison experiment (experimental details

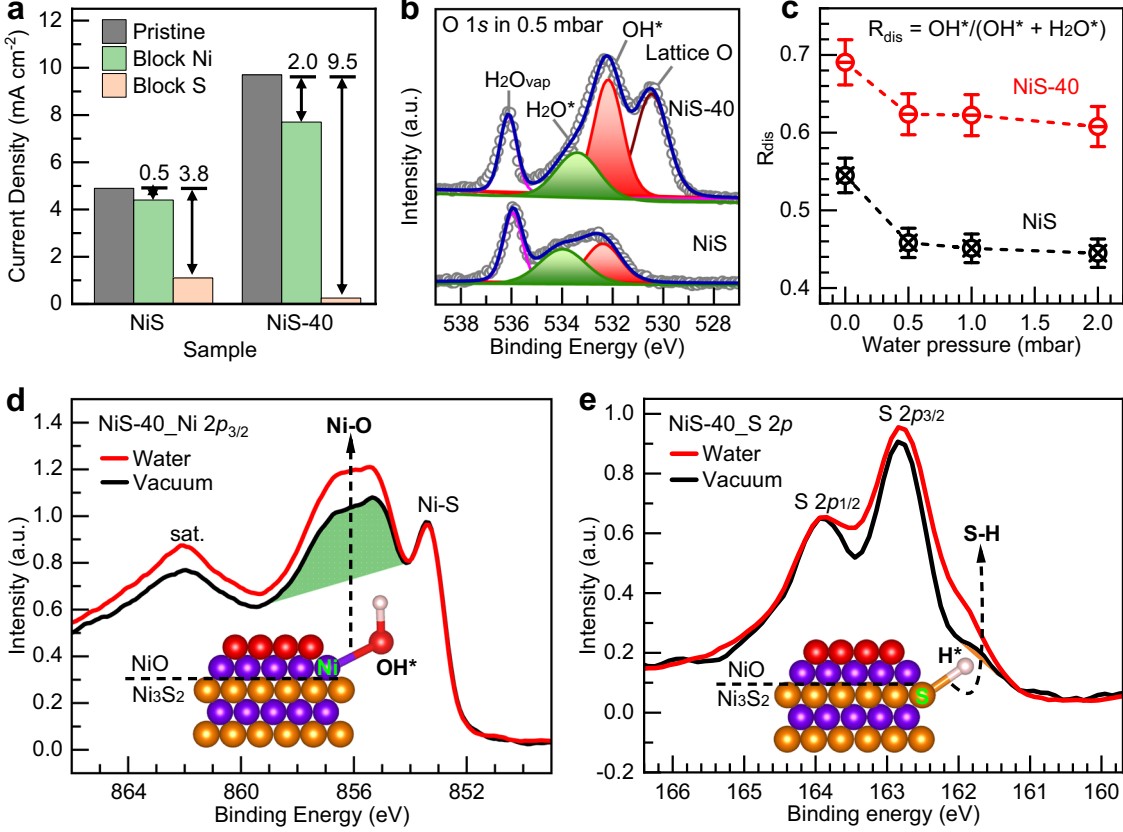

**Fig. 4 | Identification of the active sites by poisoning and NAP-XPS experiments. a** The current density of NiS and NiS-40 (NiO/Ni$_3$S$_2$) at a constant applied potential of −0.15 V vs. RHE before and after blocking the Ni sites and S sites. **b** O $1s$ XPS spectra and their peak fitting for NiS and NiS-40 at 0.5 mbar water pressure. **c** The ratio of OH* to H$_2$O* + OH* ($R_{dis}$) as a function of different water pressure (0.0 mbar, 0.5 mbar, 1.0 mbar and 2.0 mbar). **d** The Ni $2p_{3/2}$ XPS spectra of NiS-40 in vacuum and 0.5 mbar water vapor. **e** The S $2p$ XPS spectra of NiS-40 in vacuum and 0.5 mbar water vapor.

in the Methods section). Specifically, we used thiocyanate ions (SCN⁻) and dodecanethiol to selectively block the Ni and S sites, respectively[53–55]. Figure 4a shows the change in current density at −0.15 V vs. RHE before and after blocking Ni and S sites in as-synthesized NiS and NiS-40 (Ni$_3$S$_2$/NiO). The change in current density is a measure of the activity of a specific site; essentially, the greater the change after blocking a site, the higher the catalytic activity of the site in the non-poisoned state. For as-synthesized NiS, there is only a 0.5 mA cm$^{-2}$ decrease in the HER current density after blocking Ni sites, while 3.8 mA cm$^{-2}$ after blocking S sites; suggesting that S sites are the active sites. On the other hand, for the NiS-40 (Ni$_3$S$_2$/NiO) sample, it can be seen that the change is 2.0 mA cm$^{-2}$ after blocking Ni sites, and 9.5 mA cm$^{-2}$ after blocking S sites. This result indicates that both the Ni and S sites are the active sites in Ni$_3$S$_2$/NiO.

The above experiments indicate the synergistic effect of Ni$_3$S$_2$ and NiO in the transformed phase leading to the enhanced HER activity, in which the interfacial Ni sites play a crucial role in accelerating the rate determining step of water dissociation and producing H* intermediates while the interfacial S facilitates H* adsorption to generate H$_2$.

In order to support our experimental mechanistic studies for the enhanced HER activity after the formation of the mixed phase of Ni$_3$S$_2$/NiO, we performed theoretical calculations based on density functional theory (DFT) (computational method in the Methods section). In our model, we employ a heterostructure of Ni$_3$S$_2$ and NiO with O sites covered by H (marked as Ni$_3$S$_2$/NiO). We also construct models for individual Ni$_3$S$_2$ and NiO for comparison. Schematics and details of the models are shown in Supplementary Fig. 18, and the atomic

coordinates of the optimized models are supplied in Supplementary Data 1. We used crystalline NiO with an FCC crystal structure in our model because our thermodynamic analyses suggest that the stable state of NiO is crystalline (see Supplementary Fig. 19, and corresponding atomic coordinates are supplied in Supplementary Data 1). The Ni$_3$S$_2$/NiO model with O sites covered by H can be rationalized because the adsorption of H at the unsaturated oxygen sites of NiO is spontaneous and very strong, as indicated by the large values of $\Delta G_{H^*}$ of up to −0.86 eV shown in Supplementary Fig. 20. We carried out our DFT mechanistic study based on comparisons of these three modes, drawing conclusion from the trends we found rather than from individual values.

Using these models, we calculated the Gibbs energy change toward HER for the Ni$_3$S$_2$/NiO heterostructure and the corresponding individual phases, and the results are graphically shown in Fig. 5a. For individual Ni$_3$S$_2$, the energy barrier for the water dissociation step is high (0.42 eV), which is mainly due to the weak adsorption of OH* on Ni sites (Supplementary Fig. 21). Similarly, the water dissociation energy for individual NiO catalyst is as high as 0.51 eV, due to the weak bonding strength between Ni and adsorption H*. On the other hand, for the Ni$_3$S$_2$/NiO heterostructure, by taking complementary advantages of stronger adsorption of OH* at Ni sites at the NiO side and stronger adsorption of H* at S sites at the Ni$_3$S$_2$ side, the energy barrier for water dissociation can be dramatically reduced to 0.11 eV. Furthermore, Ni$_3$S$_2$/NiO heterostructures also facilitate the coupling of H* to produce H$_2$. The adsorption energy of H* on S sites at the Ni$_3$S$_2$/NiO heterostructure is −0.18 eV, which is closer to the optimal value (0 eV) for coupling of H* to form H$_2$, compared with the corresponding value

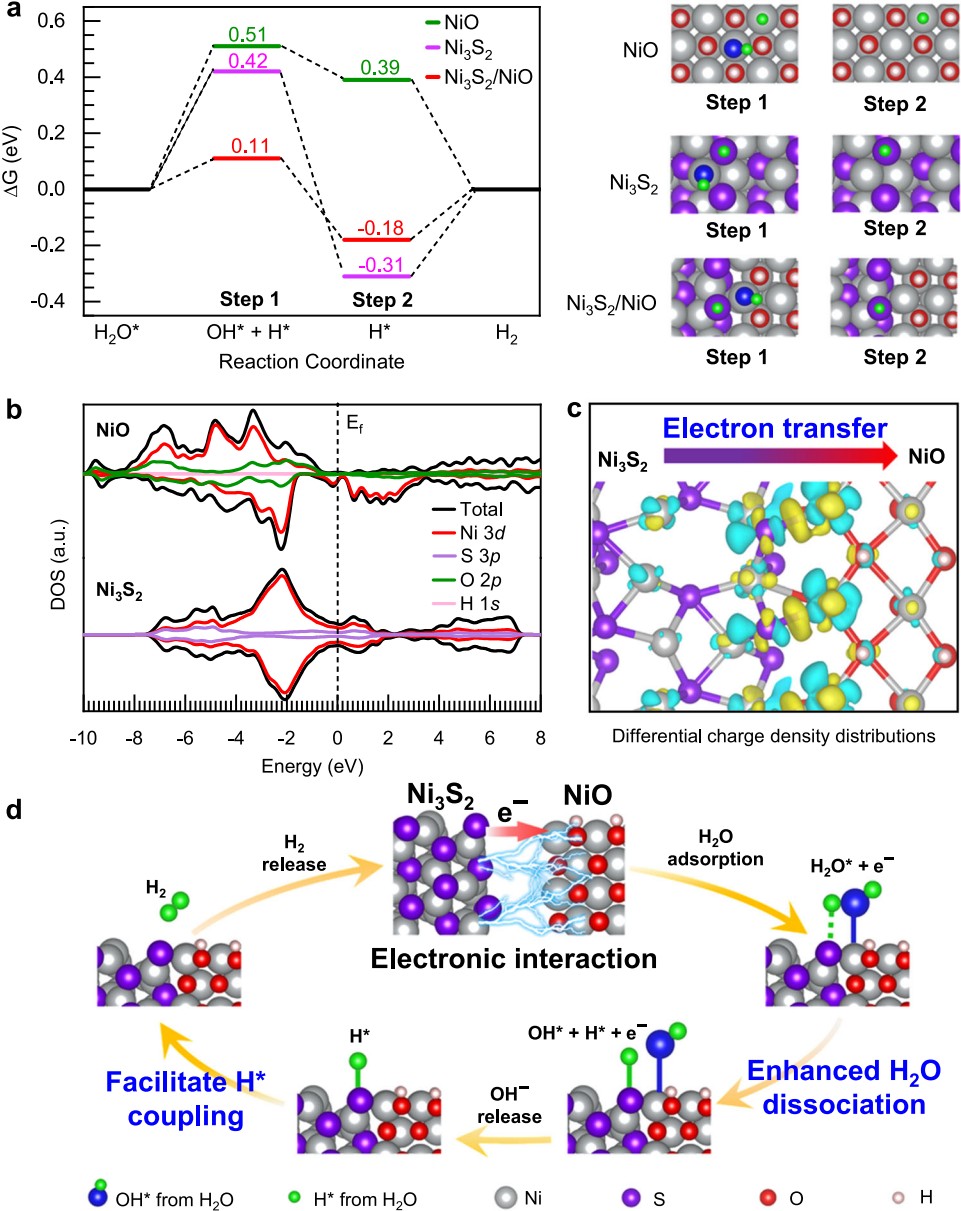

**Fig. 5 | Theoretical investigations on the HER mechanism. a** Gibbs free energy diagrams of alkaline HER pathway at Ni-Ni sites of NiO, Ni-S sites of $Ni_3S_2$ and the interfacial Ni-S sites of $Ni_3S_2/NiO$. **b** The calculated total and partial density of states of NiO and $Ni_3S_2$. **c** Differential charge density distributions of $Ni_3S_2/NiO$. Yellow contours represent electron accumulation and cyan contours represent electron depletion. **d** The schematic diagram for HER mechanism of $Ni_3S_2/NiO$.

(−0.31 eV) for individual $Ni_3S_2$ (Fig. 5a). Interestingly, our DFT results reveals that a strong electronic interaction between $Ni_3S_2$ and NiO at the interface provide favorable energetics for coupling of H* to form $H_2$. Figure 5b shows the density of states (DOS) of NiO and $Ni_3S_2$. NiO has low DOS at the Fermi level, and $Ni_3S_2$ has a metallic state, showing a high density of occupied electrons around the Fermi level. Therefore, electrons can easily transfer from $Ni_3S_2$ to the empty state of NiO. As shown in Fig. 5c, the charge differential analysis for the $Ni_3S_2/NiO$ hetero-interface clearly reveals that electrons transfer from S sites of the $Ni_3S_2$ side to the NiO side. The depletion of electrons at S sites of the $Ni_3S_2$ side can tune the electronic states of S atoms further weaken the adsorption energy of H* from −0.31 eV to −0.18 eV, which provides favorable energetics for coupling of H* to form $H_2$. Therefore, as shown in the schematic model in Fig. 5d, our experimental and theoretical mechanistic study results clearly reveal that the in-situ formed $Ni_3S_2/$NiO hetero-interface provides dual active sites to promote water dissociation, and the strong electronic interaction between $Ni_3S_2$ and NiO

interface also modifies the electronic state of interfacial S sites and thus provide optimized energetics for coupling of H* to form $H_2$.

In summary, an operando XAS and operando Raman were employed to investigate the dynamic restructuring of NiS under alkaline HER conditions. It was found that NiS was electrochemically reduced to $Ni_3S_2$, which subsequently oxidized with water and $OH^-$ ions from the electrolyte, leading to the formation of a highly active $Ni_3S_2/NiO$ hetero-interface. The combination of operando spectroscopy and theoretical investigations further unveiled that the $Ni_3S_2/$NiO hetero-interface would function as the actual active sites, i.e., the interfacial Ni sites provide the active sites to accelerate water dissociation while the interfacial S sites facilitate the adsorption of H* intermediates and its subsequent association to form $H_2$. Our work shows that the chemistry of transition metal chalcogenides is highly dynamic, and careful control of the working conditions may lead to the in-situ formation of catalytic species that boot their catalytic performance.

## Methods

### Synthesis of NiS samples

Carbon paper was used as the substrate for NiS. Carbon paper was consecutively washed with HNO$_3$ (1 M) and deionized water under sonication for 15 minutes in each solution to thoroughly remove impurities and improve hydrophilicity. 0.2448 g nickel acetate (99 wt%, Alfa Aesar) and 0.2300 g thioacetamide (99 wt%, Alfa Aesar) were dissolved in 35 mL deionized water. After stirring, the mixture was poured into a 50 ml autoclave with a piece of carbon paper (0.5 × 1.5 cm²). The growth was carried out at 120 °C in an electric oven for 12 h. After the autoclave cooled down naturally to room temperature, the samples were removed, washed with deionized water, and then dried in a vacuum oven at room temperature.

### Preparation of electrolyte and electrode

1 M KOH can be obtained by dissolving 32.0058 g of KOH (85 wt%, Sinopharm Chemical Reagent) in 500 ml of deionized water. Store it in a plastic bottle. We usually prepare it freshly as needed. The pH value was 14.00 ± 0.03 measured by pH meter (details in the Supplementary Table 8 and Supplementary Fig. 22). As shown in Supplementary Fig. 1a, a portion of the carbon paper with 0.5 × 1.0 cm² of geometric dimensions is covered by the NiS sample, while another portion with 0.5 × 0.5 cm² of geometric dimensions is left bare for connection to the glassy carbon electrode. Clamping the exposed portion of the carbon paper with the glassy carbon electrode forms the working electrode.

### Material characterizations

The crystal structure was analyzed using X-ray diffraction (XRD) employing Cu Kα radiation. Surface morphology was assessed using a ZEISS Sigma field emission scanning electron microscope (SEM). For analysis with transmission electron microscopy (TEM) and spherical aberration-corrected transmission electron microscopy (SAC-TEM), samples were suspended in absolute ethanol and subsequently deposited onto a Cu grid. High-resolution X-ray photoelectron spectroscopy (XPS) measurements were conducted using a monochromatic Al Kα1 X-ray source ($hv$ = 1486.6 eV) equipped with a SPECS PHOIBOS 150 electron energy analyzer, achieving a total energy resolution of 0.50 eV. The binding energy was calibrated using the C 1$s$ peak[34].

### Electrochemical measurements

All electrochemical measurements were carried out using a three-electrode configuration controlled by a CHI 750E electrochemical workstation, consisting of an as-prepared sample on carbon paper as working electrode, Hg/HgO (1 M KOH) as the reference electrode and graphite rode as the counter electrode (Supplementary Fig. 23). The reference electrode was calibrated by comparing it with the standard Hg/HgO (1 M KOH) purchased from Tianjin Aida Co., LTD. The potentials reported in this work were normalized versus the RHE using $E_{RHE} = E_{Hg/HgO} + 0.059$ pH + 0.098 V. We conducted all HER measurements in 1 M KOH electrolyte. Prior to measurements, all fresh electrolytes were bubbled with pure nitrogen for 30 minutes. Linear sweep voltammetry (LSV) was employed to obtained polarization curves by sweeping the potential from 0.00 V to –1.00 V vs. RHE at a scan rate of 5 mV s$^{-1}$. The electrochemical impedance spectroscopy (EIS) was performed in the same configuration at –0.20 V vs. RHE applied potential over frequency range from 100 kHz to 0.1 Hz at the amplitude of the sinusoidal voltage of 5 mV. To correct the ohmic drop, the measured potentials were calibrated using equation $E_{cor} = E - iR$, where $E_{or}$ was corrected potentials, E was measured potential, $i$ was current and $R$ was the contact resistance determined either through $iR$ measurement or by analyzing EIS curves. In this work, 85% $iR$ was conducted. The polarization curves were replotted as overpotential (η) versus log current (log $j$) to get Tafel plots for assessing the HER kinetics of investigated catalysts.

Capacitance measurements in the potential region of no faradaic process at different scan rates of 20, 40, 60, 80, 100, and 120 mV s$^{-1}$ can be used to determine electrochemical active surface area (ECSA).

### Operando electrochemical X-ray absorption experiment

Operando XAS measurements were carried out on NiS samples grown on carbon paper as the working electrode at the CLAESS beamline of the ALBA synchrotron in Spain, using a specially designed in-house electrochemical cell setup (see Supplementary Fig. 24). A platinum wire served as the counter electrode, and an Ag/AgCl (3 M KCl) electrode was used as the reference electrode in a 1 M KOH solution. These electrochemical experiments were done in a computer-controlled electrochemical workstation. Spectra at the Ni K-edge were continuously collected as the samples were subject to cyclic voltammetry (CV) from the open circuit potential to –1.50 V versus RHE and scan rate of 0.5 mV s$^{-1}$. Spectra acquisition was performed using a Si (311) monochromator that offered an incident energy resolution of 0.3 eV. Appropriate angle and coating for the collimating and focusing mirrors were selected to ensure harmonic rejection. The measurements were done in fluorescence mode, and both NiO pellets and Ni foils were employed for energy calibration. XAS data processing was conducted using the ATHENA software package[56].

### Operando Raman experiment

Raman spectra were acquired using a Jobin-Yvon Horiba Xplora confocal Raman system. All experiments were done with an excitation wavelength of 638 nm and a ×50 microscope objective with a numerical aperture of 0.55. Laser power was maintained at -1.5 mW. Operando experiments took place in a K006 Raman cell (see Supplementary Fig. 25), and Raman spectra were recorded at various applied potentials.

### Near-ambient-pressure photoemission spectra experiment

Near-ambient-pressure photoemission spectra (NAP-XPS) were recorded on a lab-based spectrometer (SPECS GmbH, Berlin) using a monochromated Al Kα source ($hv$ = 1486.6 eV) operating at 50 W. The analyzer is a SPECS PHOIBOS 150 NAP, a 180° hemispherical energy analyzer with a 150 mm mean radius. The entrance to the analyzer is a nozzle with a 300 μm diameter orifice. The analyzer is run in fixed analyzer transmission (FAT) mode. The pass energy was set to 40 eV for survey scans and 20 eV for high-resolution regions. Water vapor dosing was carried out via a leak valve. Ni 2$p$, S 2$p$, and O 1$s$ NAP-XPS at all pressures were recorded after at least 15 minutes of pressure stabilization.

### Poisoning experiment

In order to explore what is the active site during the HER process, the influence of thiocyanate ion (SCN$^-$) and dodecanethiol on the HER activity of investigated catalysts was evaluated by adding 10 mM SCN$^-$ and 10 mM dodecanethiol in the electrolyte respectively, where SCN$^-$ and dodecanethiol are known to poison the Ni sites and S sites, respectively[53–55].

### Density functional theory calculations

For the density functional theory calculations (DFT), Perdew-Burke-Ernzenhof (PBE) method is used to obtain exchange correlation functional[57], and projector-augmented wave (PAW) pseudopotentials are employed[58,59]. Hubbard-U and DFT-D3 (B-J) are also introduced in the calculations to correctthe electron correlation. The whole DFT calculations are implemented in the Vienna Ab initio Simulation Package (VASP) code[60,61]. The value of U-J term was set as 4.0 eV for Ni cations[62,63]. A Γ-centered Monkhorst-Pack grid with a 2 × 2 × 1 k-point mesh was used. The wave functions of valence electrons were

expanded using a plane wave basis set with a cut-off energy of 500 eV. The convergence criterion was set as $10^{-6}$ eV between for electronic optimization. For the structural optimization loops, the maximum force on each atom was less than $-0.01$ eV/Å[64]. All calculations were spin-polarized.

## Gibbs free energy calculations

The steps involved in HER in alkaline media are listed in Eqs. (1)–(3), as shown below:

$$\text{Volmer}: H_2O\,(l) + 2M^* \rightarrow M^*OH + M^*H \tag{1}$$

$$M^*OH + M^*H + e^- \rightarrow M^* + M^*H + OH^- \tag{2}$$

$$\text{Heyrovsky}: M^*H + H_2O + e^- \rightarrow M^* + H_2 + OH^- / \text{Tafel}: M^*H \rightarrow M^* + 1/2H_2 \tag{3}$$

Here, we applied a method developed previously for modeling the thermochemistry of electrochemical reactions based on density functional calculations[64]. Specifically, the Gibbs free energy of reactions, served as the criterion to determine the spontaneity of reactions, can be acquired based on DFT with corrections including entropic (TS) and zero-point energy (ZPE) contributions by the following equation:

$$\Delta G = \Delta E_{DFT} + ZPE - T\Delta S \tag{4}$$

Here, the $E_{DFT}$ represents the DFT energy computed for the respective systems, ZPE refers to the change in zero-point energy determined from the vibrational frequencies, and $\Delta S$ is the change in the entropy. For Gibbs free energy calculations, the free energy of a proton ($G_{H^+}$) is replaced by half of a hydrogen molecule in gaseous $H_2$ ($1/2G_{H_2}$) for every step of electrochemical reactions[3]. The entropies of corresponding species were obtained from the vibrational frequencies. For gas molecules such as $H_2$, the value of entropy can be described in the sum of the translational, vibrational, and rotational sections. Furthermore, the translational and rotational entropic values were not considered due to negligible contributions. The entropy is obtained by the following equation:

$$S(T) = \sum_{i=1}^{3N} \left[ -R\ln\left(1 - e^{-\frac{h\upsilon_i}{k_BT}}\right) + \frac{N_A h\upsilon_i}{T} \frac{e^{-h\upsilon_i/k_BT}}{1 - e^{-h\upsilon_i/k_BT}} \right] \tag{5}$$

where $R$ denotes the gas constant, $k_B$ is the Boltzmann constant, $h$ represents Plank's constant, $N_A$ corresponds to Avogadro's number, $\upsilon_i$ signifies the frequency and $N$ is the number of adsorbed atoms. Additionally, for frequency calculations, all substrate atoms were fixed and only adducts-related reaction intermediates are calculated, and the contributions from the catalyst phase to ZPE and $\Delta S$ are not involved in the Gibbs free energy of those reactions. The accuracy of calculations is provided in Supplementary Note 6.

## Data availability

Additional data supporting this study's findings are available from the corresponding author upon request. Source data are provided with this paper.

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

## Acknowledgements

The authors acknowledge funding supports from National Natural Science Foundation of China under Grant Nos. 22075232, 22275154, and 52225104. K.H.L.Z., J.P.H., and X.D. also acknowledge the Mobility Program of the Sino-German Center for Research Promotion (Grant No. M-0377). R.W. is grateful for funding support by "Shuguang Program" supported by the Shanghai Education Development Foundation and Shanghai Municipal Education Commission (No. 20SG03), and the Science and Technology Commission of Shanghai Municipality (No. 22520710600). F.E.O. thanks to MINECO and European NextGenerationEU/PRTR Fund for the Ramón y Cajal contract (RyC2021-034254-I).

V.A.P.O. is grateful for the funding supported by the EU (HYSOLCHEM project with grant agreement No. 101017928) and Spanish AEI (SOLFuture PLEC2021-007906). Giulio Gorni acknowledges Grant FJC2020-044866-I funded by MCIN/AEI/ 10.13039/501100011033, by Plan de recuperación, transformación y resiliencia, and by the "European Union NextGenerationEU/PRTR". Mariam Barawi acknowledges NovaCO2 project (PID2020-118593RB-C22) and the RYC2022-038157-I Grant funded by MCIN/AEI/ 10.13039/501100011033. M.G.-T. acknowledges PEC2Change project (TED2021-129999A-C33) funded by MCIN/AEI/ 10.13039/501100011033 and the European Union Next Generation EU/ PRTR. Additionally, the project that gave rise to these results received the support of a fellowship from "la Caixa" Foundation (ID 100010434). The fellowship code is LCF/BQ/PR23/11980046. Part of these experiments was performed at the CLAESS beamline of the ALBA synchrotron (proposal number AV-2021035069) with the help of the ALBA staff. The researchers would like to thank Iván García Dominguez and Francisco Martínez López, floor coordinators at ALBA Synchrotron, for their help with the design and 3D printing of the electrochemical cell used for in situ X-ray absorption measurements.

## Author contributions

K.H.L.Z., R.W., and X.D. designed and supervised the research. X.D. and P.Z. prepared the samples. X.D. carried out electrochemical measurements and structure characterization, and analyzed the data. D.L. accomplished the DFT calculations. X.D., F.E.O., G.G., X.C., and V.A.P.O. contributed to all operando experiments and data analysis. D.L., H.W., M.B., M.G.-T., J.P.H., J.L., J.K., and S.C. participated in the discussion of the data. K.H.L.Z., R.W., and F.E.O. revised the manuscript. X.D. and D.L. wrote the manuscript. All authors contributed to the final version.

## Competing interests

The authors declare no competing interests.
