## [Peer Review File · Nature Communications]

Dynamic restructuring of nickel sulfides for electrocatalytic hydrogen evolution reactionReviewer #1 (Remarks to the Author):

This report is regarding in situ formed NiO_x/Ni₃S₂ for HER and their operando study into the active sites and phase transitions. There is a lack of novelty for a high IF journal like Nat Comm since related works have already been demonstrated for other similar HER catalysts like NiS₂, CoS₂, NiSe₂, as pointed out in ref 37 to 39. In my opinion, this work does not meet the high novelty standards of Nat Comm. Recommend a more specialized journal like JMCA.

Reviewer #2 (Remarks to the Author):

The paper deals with a study of NiS-based electrocatalyst for the hydrogen evolution reaction (HER) in alkaline media.

The authors propose that during operation, NiS transforms into a mixture of Ni₃S₂ and NiO_x phases, the interface of which provides active centers of HER activity.

The phase conversion process was followed mainly using operando Ni K-edge X-ray absorption spectroscopy (XAS). Several other experimental techniques as XRD, SEM, TEM, XPS, Raman spectroscopy were used as well. The theoretical support was provided by DFT calculations.

The obtained results are interesting and are based on several nice experiments. However, there are many points that should be clarified and/or explained in more detail. Some conclusions are not supported by the results. Therefore, I cannot recommend the manuscript for publication in the present form.

The authors should consider the following points:

1) It is not clear how were obtained the reference samples Ni₃S₂ and NiO_x/NiO?

2) What is the meaning of NiO_x phase? What is the difference between NiO and NiO_x phases?

3) The linear combination analysis of the Ni K-edge XAS spectra based on reference data for Ni₃S₂ and NiO compounds should be preceded by the principal component analysis to prove the number of phases present in the sample. The use of the crystalline NiO phase to simulate the contribution from some NiO_x phase should be reasoned.

4) The Ni K-edge XANES data in Fig. 2(c) (region II and III) are rather poor in features and short in the energy range, also the fits in Figs. 2(c) and S6 (dashed lines) are not ideal. Does it mean that other phase or phases can be present in the sample? Can authors perform the same analysis using the Ni K-edge EXAFS spectra in larger energy or k-space?

5) The description of the EXAFS simulations/fits is fully missing. This makes unclear the origin of data in Tables S3 and S4. The description of the simulation procedure and the approximations used should be given in the Supplemental Materials.

6) The reliability of the fits is also not clear. Some values reported in the two tables look unphysical. In particular, all Ni-O distances are very short being about 1.7 Å (a longer distance of 2.01 Å is reported in the text!), whereas the Ni-O bond length in bulk NiO is significantly longer being about 2.09 Å.

7) The fitting results should be provided (in the Supplemental Materials) for data reported in Fig. 2(d).

8) The Ni K-edge EXAFS spectra should be additionally reported in Fig. 3, and the Ni-O and Ni-S contributions should be shown separately. Currently, the contribution from the Ni-S bonds is not convincing – it could be negligible.

9) The comparison between the experimental and fitted EXAFS spectra should be provided in separate figures for the results reported in Tables S3 and S4 in the Supplemental Materials as well

as in Fig. 3(b).

10) The values of coordination numbers (CNs) should contain errors. The meaning of asterisks should be explained in the captions of Tables. The small values of the Ni-O CNs do not convince for the occurrence of the NiO phase.

11) It looks like Fig. 3(d) contains wrong labels for the spectra: the blue line, probably, corresponds to NiO. One can recommend using the same colors in Fig. 3(b) and (d).

12) The Raman data (Fig. 2e) are available in a short-range, so only two bands around 320 and 480 cm^{-1} are visible in the catalyst. Their assignment cannot be unique. Also, the band at 560 cm^{-1} discussed in the text and attributed to NiOx phase is not present in the Raman spectra. Therefore, the Raman data should be re-analysed. Comparison with the literature data should be done.

13) It is difficult to judge the accuracy of DFT calculations since no comparison with any reference data is provided. It is not clear how well the structural and electronic properties are reproduced, in particular, taking into account that the PBE functional is not hybrid. The position of the Fermi level should be given in Fig. 5(d).

Reviewer #3 (Remarks to the Author):

Report on "In situ formed NiOx/Ni₃S₂ Nano interfaces Boost High Hydrogen Evolution Reaction Activity in Alkaline Conditions" by X. Ding, et. al

The authors report the results of a comprehensive experimental and theoretical study exploring the structural transformations that occurring within NiS and which accelerate the hydrogen evolution reaction (HER) under alkaline conditions. This is an important area of work with much interest in the formation of H₂ as an energy storage medium through the electrolysis of water using renewable energy sources. The results presented by the authors suggest that NiS undergoes a phase transformation during the reaction to form an NiOx/Ni₃S₂ phase, which then acts as a catalyst for the rate-limited step in the HER under alkaline conditions. This is an important observations because understanding the structure on which a reaction occurs is key for the design of improved catalysts. As such, this manuscript will be of broad interest and importance. I recommend publication after the following points have been adequately addressed. Note that given that my expertise lies in theoretical chemistry, I have only provided comments related to the calculations report by the authors.

1. The theoretical methods section in the supporting information do not adequately describe key parts of the calculations. In particular, the authors should describe what kinds of pseudopotentials were used (I assume PAWs, but it isn't stated), what planewave cutoff was used for the density and/or augmentation charges, and the k-point grid used in the calculation. In addition, the authors should outline how zero-point energies and entropies were obtained. I assume through phonon calculations, but it is not stated. It is also important to state whether such calculations (if they were phonon calculations) were performed for the whole system or just in the region near the reactive site.

2. In the first reaction step in Figure 3a, the authors show S leaching from the material and indicate that oxygen should fill the vacancy. However, the middle image in that figure states that the material is Ni₃S₂ and shows no oxygen. So, how does oxygen factor in? I found this pretty confusing.

3. In Figure 5c, the authors show the structure of NiOx/Ni₃S₂ and suggest a mechanism. The mechanism relies on water binding through the oxygen atom to a nickel atom that is bonded to an oxygen and a sulfur. Based on the structure provided, it's not clear that a site facilitate this process

is present and the images provided in the supporting information (Figures S9 - S11) don't really support this mechanism, either. It would be helpful if the authors indicated more accurately in the main text how this reaction occurs. In addition, the authors should provide the coordinates of all calculated structure to the supporting information so others can easily reproduce the results.

4. The mechanism outlined in the bottom of Figure 5c involves a proton transferring to sulfur. It seems that transfer to oxygen would be preferred. Perhaps I missed the explanation of why this is the case. However, it is important to explain why the oxygen atom isn't protonated, since most people would intuitively expect that to be preferred.

RESPONSE TO REVIEWERS' COMMENTS

20th July 2022

Dear Editor,

We address all reviewers' comments point-by-point in the following.

Referee: 1

Comments to the Author:

This report is regarding in situ formed NiO_x/Ni₃S₂ for HER and their operando study into the active sites and phase transitions. There is a lack of novelty for a high IF journal like Nat Comm since related works have already been demonstrated for other similar HER catalysts like NiS₂, CoS₂, NiSe₂, as pointed out in ref 37 to 39. In my opinion, this work does not meet the high novelty standards of Nat Comm. Recommend a more specialized journal like JMCA.

Response – We understand the view of referee 1 that the dynamic nature of metal sulfides chemistry has already been demonstrated, and therefore, transitions under an electrochemical reaction are rather expected. However, the observation and mechanistic elucidation of the phase transition of NiS to Ni₃S₂ is only a part of the work we report in our paper. The identification of the “real” active sites under HER operation conditions and the HER mechanism of the transformed NiS electrodes are the most important discoveries and define the novelty of our work. We found the HER activity does not only depend on the conversion of NiS to Ni₃S₂, but also on the formation of surface NiO. We show how the interaction of active sites on NiO and Ni₃S₂ may lead to the observed high catalytic performance.

We strongly believe these findings are appealing to the broad readership of *Nature Communications* because, (1) we have shown how the dynamic chemistry of metal

sulfides, which are materials with multiple catalytic applications, can be controlled to promote the formation of highly active catalytic species; (2) we describe a very interesting dual-site catalytic system, and this type of systems provide alternative reaction mechanisms that may break the scaling relationships in the formation energy of intermediates that often limits the catalysis of not only the HER but also the OER and the ORR, offering more chances for achieving better catalytic performance compared with single active site catalysts; (3) our experimental methodology, combining *in-situ* and *operando* spectroscopy, experimental mechanistic studies and DFT calculations may be applied for the study of many other challenging metal chalcogenide systems.

We have included remarks on the significance of our findings **highlighted in green** in the introduction and in the discussion, linking them to other works that can benefit from our observations.

Referee: 2

Comments to the Author:

The paper deals with a study of NiS-based electrocatalyst for the hydrogen evolution reaction (HER) in alkaline media.

The authors propose that during operation, NiS transforms into a mixture of Ni₃S₂ and NiO_x phases, the interface of which provides active centers of HER activity.

The phase conversion process was followed mainly using operando Ni K-edge X-ray absorption spectroscopy (XAS). Several other experimental techniques as XRD, SEM, TEM, XPS, Raman spectroscopy were used as well. The theoretical support was provided by DFT calculations.

The obtained results are interesting and are based on several nice experiments. However, there are many points that should be clarified and/or explained in more detail. Some conclusions are not supported by the results. Therefore, I cannot recommend the manuscript for publication in the present form.

The authors should consider the following points:

Response – We thank referee 2 for his/her positive comments on our work. We are appreciated for the helpful suggestions/comments for improving our manuscript. Followings are our responses to the comments.

[The comments are shown in *italic*; responses are in black; all revisions in manuscript and supplementary information are highlighted in yellow.]

Comment 1. *It is not clear how were obtained the reference samples Ni₃S₂ and NiO_x/NiO?*

Response 1 – The reference sample for NiO was a commercial powder. In order to get the Ni K-edge XAS reference of NiO, proper amount of NiO sample was weighted, mixed with cellulose and then pressed into a 13 mm pellet for XAS measurement in transmission mode. For Ni₃S₂ reference, it was a crystal powder synthesized by ourselves (the synthesis procedure and the characterization are shown below). In order to get the Ni K-edge XAS reference of Ni₃S₂, proper amount of Ni₃S₂ powder was weighted, mixed with cellulose, and then pressed into a 13 mm pellet for XAS measurement in transmission mode. The spectrum was aligned with our data by a comparison with a Ni foil. (We add this descriptions in Supplementary Note 2 in page 6 of the revised Supplementary Information)

The Ni₃S₂ reference was synthesized by heating stoichiometrically mixed elements of nickel and sulfur at 800 °C for 6 hours in silica tube sealed under vacuum. The XRD patterns shown in Fig. R1a confirm that as-synthesized Ni₃S₂ powder has high crystalline quality. The shapes and positions of Ni 2p and S 2p XPS spectra shown in Figs. R1b-c are also the same to previous report of Ni₃S₂¹. As shown in Fig. R1d, the Ni K-edge spectrum for our Ni₃S₂ reference also agree well with the spectrum from the Farrel-Lytle database (http://ixs.iit.edu/database/data/Farrel_Lytle_data/), which allows us to confidently use the Ni₃S₂ reference for our analysis. (We add this descriptions in Supplementary Note 1 in page 5 of the revised Supplementary Information.)

Fig. R1. (a) XRD patterns of the as-synthesized Ni₃S₂ crystal powder and the standard card PDF#85-1802. (b) Ni 2p XPS spectra and (c) S 2p XPS for as-synthesized Ni₃S₂. (d) Ni K-edge XAS spectra of as-synthesized Ni₃S₂ and Ni₃S₂ spectra from the Farrel-Lytle database (http://ixs.iit.edu/database/data/Farrel_Lytle_data/).

Reference:

1. Buckley, A. N. & Woods, R. Electrochemical and XPS studies of the surface oxidation of synthetic heazlewoodite (Ni₃S₂). *J. Appl. Electrochem.* **21**, 575-582 (1991).

Comment 2. What is the meaning of NiO_x phase? What is the difference between NiO and NiO_x phases?

Response 2 – We used the term “NiO_x”, because we do observe the formation of nickel oxide, but we are not able to determine the crystal structure or the phase of the nickel

oxide, because no XRD pattern was observed. However, from our XPS (Fig. R2a), XAS (Fig. R2b) and Raman spectroscopy (Fig. R2c) measurements, we found that the spectroscopic data fit well with NiO. Therefore, we can confirm that the nickel oxide should be NiO. Since no XRD peak was observed (Fig. R2d), the NiO should be in form of small amorphous clusters or thin layers.

We changed “NiO_x” to “NiO” in the revised manuscript. We also discuss the structure of NiO in page 7 in the revised manuscript.

Fig. R2. (a) the Ni 2p XPS spectrum for NiS after 40-minute HER measurements (marked as NiS-40), along with NiO and Ni₃S₂ reference. That indicates the mixture contains NiO and Ni₃S₂. (b) The Ni K-edge XAS spectra at specific applied voltage of +1.00 V vs. RHE in positive sweep of CV, and their linear combination of NiO and Ni₃S₂. That also indicates the mixture contains NiO and Ni₃S₂. (c) *Operando* Raman spectroscopy of NiS at specific applied voltage of -0.30 V vs. RHE during HER process. The peaks marked as a, b, c and d stand for Ni₃S₂, and the peaks marked as e, f, g, h and i stand for NiO. (d) XRD pattern for NiS after HER measurement. Based on analysis above, the nickel oxide phase is NiO in form of small amorphous cluster or thin layers.

Comment 3. The linear combination analysis of the Ni K-edge XAS spectra based on reference data for Ni₃S₂ and NiO compounds should be preceded by the principal

component analysed to prove the number of phases present in the sample. The use of the crystalline NiO phase to simulate the contribution from some NiOx phase should be reasoned.

Response 3 – Principal component analysis (PCA) was performed using a set of 6 Ni-K edge spectra measured during the positive sweep potential. Despite PCA is a mathematical tool that cannot explain which phases are present in a certain sample, it can give an estimation of the number of phases present. (We added the PCA in Supplementary Note 3 in page 7 of the revised Supplementary Information.)

Fig. R3. Principal component analysis (PCA) of the XAS spectra measured during the positive sweep potential at six different values. The first component (mean) was multiplied by 0.5 for a better comparison.

In our case, as shown in Fig. R3, we can see that most of the spectrum is reproduced by comp. 1 (mean spectrum). Additionally, comp. 2 is necessary to explain the different spectroscopic features observed. Comp. 3 is not giving a relevant improvement of the fit and the cumulative variance is 0.999 after considering only the first two components. The other components are mainly associated with noise. On these bases, we think that the use of two references, those of Ni₃S₂ and NiO, is reasonable to explain the spectral features in our systems. (We added the PCA in Supplementary Note 3 in page 7 of the revised Supplementary Information.)

Based on PCA, the use of two references is reasonable to explain the spectral features in our system. We have followed reason to believe the two references should be Ni₃S₂ and NiO for linear combination fitting (LCF). (We added the description in Supplementary Note 4 in page 7 of the revised Supplementary Information.)

Fig. R4. (a) The Ni *K*-edge XAS spectra at specific applied voltages of +1.00 V (1 V_N means +1.00 V in negative sweep of CV), -1.50 V (-1.50 V_N means -1.50 V in negative sweep of CV), and +0.50 V, +1.00 V (+0.50 V_P and +1.00 V_P mean +0.50 V and +1.00 V in positive sweep of CV) in region I, II and III respectively, and their linear combination of Ni₃S₂ and NiO. (b) XRD pattern for NiS after HER measurement. The pattern was assigned to Ni₃S₂. (c) Ni *K*-edge XAS of amorphous NiO (a-NiO NSs) and crystal NiO (c-NiO NSs). It was extracted from previous report by R. Li *et al.*². It shows the XANES features of amorphous NiO are very similar to those of crystalline NiO.

Ni *K*-edge XAS during a cycle of CV give the adequate reasons for using Ni₃S₂ and NiO as reference. As shown region II in Fig. R4a, similarity of the measured spectra with Ni₃S₂ makes straightforward its use as a reference for the linear combination fitting. When it goes into region III in Fig. R4a, NiO was a reasonable reference because the increase of the whiteline of 8350 eV is a typical feature of NiO and the slightly shift of the edge towards higher energy indicates the presence of a more electronegative ion. Meanwhile, the XRD of NiS after HER measurement corresponds to Ni₃S₂ (Fig. R4b), which indicate it also contains Ni₃S₂ in region III. Moreover, the first shell suffers an appreciable contraction for specific potentials due to the Ni-O bond formation (Supplementary Figs. 12, 14 and Supplementary Table 4 in supplementary information). We also detect the formation of NiO by other characterizations (*e.g.*, Raman spectroscopy, XPS and TEM,

mentioned in main text). As shown in Fig. R4b, because no XRD pattern was observed for NiO, we think the NiO should be in form of small amorphous clusters or thin layers. However, the XANES features of amorphous NiO are very similar to those of crystalline NiO. As reported by R. Li *et al.*², the XANES for amorphous NiO and crystalline NiO are quite similar (Fig. R4c). For all these reasons, we think that a relevant description of the system may be obtained using Ni₃S₂ and NiO references. (We added the reasons of using Ni₃S₂ and NiO references in Supplementary Note 4 in page 8, 9 of the revised Supplementary Information.)

Reference:

2. Li, R. *et al.* Short-range order in amorphous nickel oxide nanosheets enables selective and efficient electrochemical hydrogen peroxide production. *Cell Reports Physical Science* **3**, 100788 (2022).

Comment 4. The Ni K-edge XANES data in Fig. 2(c) (region II and III) are rather poor in features and short in the energy range, also the fits in Figs. 2(c) and S6 (dashed lines) are not ideal. Does it mean that other phase or phases can be present in the sample? Can authors perform the same analysis using the Ni K-edge EXAFS spectra in larger energy or k-space?

Response 4 – As motioned response 3 to comment, based on the principal component analysis (PCA) that we introduce in the revised version of the paper, we think that the use of two references, Ni₃S₂ and NiO, is reasonable to explain the spectral features in our system. However, LCFs in k space may not be of good quality because shrinks/dilation of bond distances can not be taken into account in a LCF, resulting in the non-ideality of the fittings. It is also important to keep in mind that spectra are taken in a dynamic system that might be changing during the acquisition of each spectrum. We analysed spectra at potentials at which the system is more stable (right after each change in the system); however minor changes might still be happening, adding to deviations from ideality when trying to fit the data.

We have extended the LCFs in the revised manuscript, yet focused the discussion on the XANES region of the spectra. We have also performed a detailed analysis of the Ni K-edge EXAFS in the k-space. See Supplementary Figs. 13, 14, and associated Supplementary Tables 3, 4 in page 18-20 of the revised Supplementary Information.

Comment 5. The description of the EXAFS simulations/fits is fully missing. This makes unclear the origin of data in Tables S3 and S4. The description of the simulation procedure and the approximations used should be given in the Supplemental Materials.

Response 5 – We have reworked on the EXAFS fits and added the following description for the simulation procedure in Supplementary Note 5 in page 9 of the revised Supplementary Information, and more description for Supplementary Table 3 (Table R1) and Supplementary Table 4 (Table R2) in Supplementary Discussion 2, 3 in page 19, 20 of the revised Supplementary Information.

“The EXAFS data were analysed in the k-range 3-11 Å⁻¹ and the Fourier Transform performed using a Hanning window. The data were fitted in the R-range 1-2.6 Å using a multiple k-weight fitting. The fits were performed using the most intense scattering paths Ni-S and Ni-Ni associated to Ni₃S₂ and NiS. When there were evidences of the presence of Ni-O bonds, the single scattering path of NiO was added. We weighted the Ni-S and Ni-O by introducing a parameter x. The amplitude of Ni-S and Ni-O scattering paths was x and 1-x, respectively. We left x as a free parameter of the fit.”

Table R1 *Operando* EXAFS fitting parameters at the Ni-K edge of NiS at different applied voltages during the stage of negative sweep of CV. ($S_0^2 = 0.8$)

Applied voltage (V vs. RHE)	Path	CN	R (Å)	σ^2 (Å ²)	R-factor
+1.00 (region I)	Ni-S	5.3±0.8	2.344±0.018	0.005±0.003	0.003
	Ni-Ni	2.00*	2.70±0.03	0.009±0.004	
+0.50 (region I)	Ni-S	6±1	2.352±0.017	0.009±0.003	0.007
	Ni-Ni	2.00*	2.68±0.03	0.009±0.004	
0.00 (region I)	Ni-S	6±1	2.35±0.02	0.008±0.003	0.003
	Ni-Ni	2.00*	2.68±0.02	0.005±0.003	
-0.50 (region II)	Ni-S	4.3±0.8	2.271±0.017	0.007±0.002	0.002
	Ni-Ni	4.00*	2.51±0.02	0.007±0.004	
-1.00 (region II)	Ni-S	4.3±0.9	2.28±0.02	0.005±0.001	0.005
	Ni-Ni	4.00*	2.505±0.015	0.010±0.002	
-1.50 (region II)	Ni-S	3.3±0.7	2.26±0.02	0.005±0.002	0.003
	Ni-Ni	4.00*	2.507±0.013	0.007±0.002	

CN: coordination number. **R:** bond distance. **σ^2 :** Debye-Waller factors. **R factor:** goodness of fit. The asterisks (*) mean that parameter has been kept fixed during the fitting process.

Below Table R1 (Supplementary Table 3), we added Supplementary Discussion 2: “For the negative sweep potential, see Table R1 (Supplementary Table 3), the results indicate the phase change from NiS to Ni₃S₂. At positive potentials, the Ni-S coordination is consistent with 6 and the bond distance of Ni-S is around 2.35 Å. The contribution from Ni-Ni scattering is also observed around 2.68 Å. These values are in good agreement with the structure of NiS. At negative potentials, the Ni-S coordination number is consistent with 4 and the bond length is around 2.27 Å, in agreement with the structure of Ni₃S₂. The Ni-Ni shell is also 4-coordinated and around 2.51 Å, similar to Ni₃S₂.”

Table R2. *Operando* EXAFS fitting parameters at the Ni-K edge of NiS in different applied voltage during the stage of *positive sweep of CV*. ($S_0^2 = 0.8$)

Applied voltage (V vs. RHE)	Path	CN	R (Å)	σ^2 (Å ²)	R-factor
-1.50 (region II)	Ni-S	3.3±0.6	2.262±0.013	0.005±0.003	0.003
	Ni-Ni	4.00*	2.50±0.01	0.007±0.001	
-1.00 (region II)	Ni-S	2.9±0.8	2.262±0.016	0.005±0.004	0.005
	Ni-Ni	4.00*	2.502±0.012	0.007±0.001	
-0.50 (region II)	Ni-S	2.9±0.6	2.258±0.012	0.006±0.001	0.002
	Ni-Ni	4.00*	2.495±0.008	0.007±0.001	
0.00 (region II)	Ni-S	3.0±0.6	2.258±0.017	0.006±0.003	0.006
	Ni-Ni	4.00*	2.50±0.01	0.006±0.001	
+0.50 (region III)	Ni-S	4 (x=0.84±0.10)	2.267±0.015	0.005±0.001	0.009
	Ni-Ni	4.00*	2.509±0.015	0.010±0.002	
	Ni-O	6 (1-x=0.15)	2.03±0.05	0.005±0.001	
+1.00 (region III)	Ni-S	4 (x=0.74±0.10)	2.286±0.014	0.005±0.002	0.011
	Ni-Ni	4.00*	2.51±0.002	0.013±0.002	
	Ni-O	6(1-x=0.25)	2.00±0.065	0.005±0.002	

CN: coordination number. R: bond distance. σ^2 : Debye-Waller factors. R factor: goodness of fit. The asterisks (*) mean that parameter has been kept fixed during the fitting process.

Below Table R2 (Supplementary Table 4), we added Supplementary Discussion 3: “The results for the positive sweep potential, see Table R2 (Supplementary Table 4), indicate

that at more negative potentials the local structure is similar to that of Ni₃S₂ with a slightly lower coordination number of Ni-S compared to the crystallographic data, which, as discussed in the manuscript, might indicated the presence of S vacancies. The Ni-S bond distance is around 2.26 Å, which is in agreement with that of Ni₃S₂. At positive potentials, there is a clear shrink of the bond length associated to the partial formation of Ni-O bonds at around 2.00 Å. The weight of the Ni-O bond compared to Ni-S is around 15 % and 25% at +0.50 and +1.00 V vs. RHE respectively (see region III in Table R2), in agreement with the values (13% and 22%) obtained from the linear combination fitting of XANES spectra, as shown in Fig. R5.”

Fig. R5. The Ni K-edge XAS spectra at specific applied voltages of (a) +0.50 and (b) +1.00 V vs. RHE (+0.50 V_P and +1.00 V_P mean +0.50 and +1.00 V in positive sweep of CV) in III, extracted from Fig. 2c, and their linear combination of Ni₃S₂ and NiO.

Comment 6. *The reliability of the fits is also not clear. Some values reported in the two tables look unphysical. In particular, all Ni-O distances are very short being about 1.7 Å (a longer distance of 2.01 Å is reported in the text!), whereas the Ni-O bond length in bulk NiO is significantly longer being about 2.09 Å.*

Response 6 – Thanks to the referee for this comment. We realize that in our previous manuscript, the including of Ni-O path in region I and II was not justified. We have removed the Ni-O path in region I and II, only include Ni-O path in region III. As we mentioned in response 5, we reworked on the fitting and the reliability of the fitting are very good. The updated results are shown in Supplementary Table 3, 4 in page 19, 20 of the Supplementary Information (*i.e.*, Table R1 and R2 in Response 5).

We made a mistake trying to fit in traces of NiO spectral features in spectra of the sample during the negative sweep of the CV. This indeed led to unreliable results in the reported

Ni-O distances, such as values about 1.7 Å. In the revised version, the EXAFS analysis was checked again and we included or not the Ni-O contribution depending on statistical estimators and physical meaning of the parameters obtained (reliable bond distance, positive defined Debye-Waller factor (σ^2) less than 0.03, and ΔE_0 less than 10 eV). We also checked the number of free parameters, the reduced chi-square and the R-factor in order to assess if the mode with NiO introduces a real improvement. In most of cases, *i.e.*, region I and II in Fig. R6a, the introduction of Ni-O scattering path was not justified from a statistical point of view. Only for the positive sweep potential, *i.e.*, region III in Fig. R6a, we can discuss the clear formation of Ni-O bonds using XAS data. We weighted the Ni-S and Ni-O by introducing a parameter x . The amplitude of Ni-S and Ni-O scattering paths are x and $1-x$, respectively. We left x as a free parameter of the fit and the updated results of fitting are shown in Supplementary Table 3, 4 (Table R1-2). As shown in Table R3 extracted from Supplementary Table 4 (Table R2), the fit give result similar to that obtained by LCF (Fig. R6b, c).

Figure R6. (a) *Operando* Ni K-edge XAS for NiS catalyst as a function of applied voltages during a cycle of CV. Top panel shows a 3-dimensional view of Ni K-edge XAS spectra; bottom shows the 2-dimensional projection from 3-dimensional view. In region III, there were clear formation of Ni-O. The Ni K-edge XAS spectra at specific applied voltages of (b) +0.50, and (c) +1.00 V vs. RHE (+0.50 V_P and +1.00 V_P mean +0.50 and +1.00 V in positive sweep of CV) in region III, and their linear combination of Ni₃S₂ and NiO. The content of NiO is 13% and 22% at +0.50 and +1.00 V vs. RHE respectively, that is similar to the results obtained by EXAFS fitting as shown in Table R3.

Table R3. *Operando* EXAFS fitting parameters at the Ni-K edge of NiS at +0.50 and +1.00 V vs. RHE during *positive sweep of CV*. ($S_0^2 = 0.8$)

Applied voltage (V vs. RHE)	Path	CN	R (Å)	σ^2 (Å ²)	R-factor
+0.50 (region III)	Ni-S	4 ($x=0.84\pm 0.10$)	2.267 \pm 0.015	0.005 \pm 0.001	0.009
	Ni-Ni	4.00*	2.509 \pm 0.015	0.010 \pm 0.002	
	Ni-O	6 (1-x=0.15)	2.03 \pm 0.05	0.005 \pm 0.001	
+1.00 (region III)	Ni-S	4 ($x=0.74\pm 0.10$)	2.286 \pm 0.014	0.005 \pm 0.002	0.011
	Ni-Ni	4.00*	2.51 \pm 0.002	0.013 \pm 0.002	
	Ni-O	6(1-x=0.25)	2.00 \pm 0.065	0.005 \pm 0.002	

CN: coordination number. **R:** bond distance. **σ^2 :** Debye-Waller factors. **R factor:** goodness of fit. The asterisks (*) mean that parameter has been kept fixed during the fitting process.

As the reviewer mentioned, highly crystalline NiO has a characteristic Ni-O distances around 2.09 Å; however, studies of short-range order in amorphous nickel oxide have shown significant increases of Ni-O distance upon increasing the crystallinity of the system. For instance, R. Li *et al.* found the Ni-O distance increases from 2.06 Å to 2.10 Å from an amorphous to a crystalline system². Also, Y. R. Denny *et al.* found the Ni-O distance increases from 1.984 Å for amorphous NiO to 2.084 Å for crystalline phase³. Therefore, the values obtained for the Ni-O distance are in agreement with the typical distances for amorphous and/or nanocrystalline NiO (in the range 2.00-2.06 Å). This further confirms the amorphous structural nature of the NiO formed under the electrocatalytic reaction.

We included the statement in page 9 of the revised manuscript. “Highly crystalline NiO has a characteristic Ni-O distances around 2.09 Å; however, studies of short-range order in amorphous NiO have shown that the Ni-O distances for amorphous and/or nanocrystalline NiO may decrease to 2.00-2.06 Å.^{2,3} This further confirms the nickel oxide formed in our work is in form of amorphous NiO.”

References:

- Li, R. *et al.* Short-range order in amorphous nickel oxide nanosheets enables selective and efficient electrochemical hydrogen peroxide production. *Cell Reports Physical Science* **3**, 100788 (2022).
- Denny, Y. R. *et al.* Ni K-edge XAFS analysis of NiO thin film with multiple scattering theory. *Surf. Interface Anal.* **46**, 997-999 (2014).

Comment 7. *The fitting results should be provided (in the Supplemental Materials) for data reported in Fig. 2(d).*

Response 7 – Thanks for this suggestion. As mentioned in response 5, we have reworked on EXAFS fitting. Fig. 2d has been updated, and the fitting results are summarized in Supplementary Tables 3, 4 in page 19-20 of the revised Supplementary Information.

Comment 8. *The Ni K-edge EXAFS spectra should be additionally reported in Fig. 3, and the Ni-O and Ni-S contributions should be shown separately. Currently, the contribution from the Ni-S bonds is not convincing – it could be negligible.*

Response 8 – Fig. 3a in previous manuscript has been moved to Fig. 2f in the revised manuscript. Figs. 3b-d were moved to Supplementary Fig. 12 in page 17 of the revised Supplementary Information.

The Ni K-edge FT-EXAFS shown in Fig. R7a (Supplementary Fig. 12a) was obtained by the Fourier transformation of Ni K-edge EXAFS spectra at +1.00 V in region III in Fig. 2c. The Ni-O and Ni-S paths have been shown separately in FT-EXAFS (Fig. R7a, *i.e.*, Supplementary Fig. 12a) and Wavelet transform (Fig. R7d, *i.e.*, Supplementary Fig. 12d). We introduced the Ni-S and Ni-O contributions for spectra where we had a clear evidence of the presence of Ni-O (PCA analysis and LCFs mentioned in response 3). As it can be seen (please see the following Table R4), the presence of Ni-O is not negligible for two potentials and its weight around 15% and 25%, in agreement with the results from LCF of XANES spectra (please see the following Fig. R8).

Table R4. *Operando* EXAFS fitting parameters at the Ni-K edge of NiS at +0.50 and +1.00 V vs. RHE during positive sweep of CV. ($S_0^2 = 0.8$)

Applied voltage (V vs. RHE)	Path	CN	R (Å)	σ^2 (Å ²)	R-factor
+0.50 (region III)	Ni-S	4 ($x=0.84\pm 0.10$)	2.267 ± 0.015	0.005 ± 0.001	0.009
	Ni-Ni	4.00*	2.509 ± 0.015	0.010 ± 0.002	
	Ni-O	6 (1-x=15)	2.03 ± 0.05	0.005 ± 0.001	
+1.00 (region III)	Ni-S	4 ($x=0.74\pm 0.10$)	2.286 ± 0.014	0.005 ± 0.002	0.011
	Ni-Ni	4.00*	2.51 ± 0.002	0.013 ± 0.002	
	Ni-O	6(1-x=25)	2.00 ± 0.065	0.005 ± 0.002	

CN: coordination number. R: bond distance. σ^2 : Debye-Waller factors. R factor: goodness of fit.

Fig. R7. (a) The FT-EXAFS obtained by the Fourier transformation of Ni K-edge EXAFS spectra at +1.00 V in region III in Fig. 2c, and its fitting by using Ni-O, Ni-S and Ni-Ni paths. (b) The Ni K-edge XAS spectra of NiO/Ni₃S₂ at +1.00 V, along with NiO and Ni₃S₂ as references. The results indicate the oxidation state of Ni in NiO/Ni₃S₂ is between +1.33 and +2. (c) Overview Wavelet Transform (WT) of the Ni K-edge EXAFS spectra with $\eta=10$, $\sigma=1$, $3 \text{ \AA}^{-1} \leq k \leq 11 \text{ \AA}^{-1}$. (d) High-resolution WT of the first shell with $\eta=1.6$, $\sigma=1$, $5 \text{ \AA}^{-1} \leq k \leq 11 \text{ \AA}^{-1}$.

Below Fig. R7 (Supplementary Fig. 12), we have added the descriptions “According to the PCA analysis, we use Ni-O, Ni-S and Ni-Ni scattering paths to fit the data, and the results of fitting was reliable because of the reasonable parameter as shown in Table R4 (Supplementary Table 4). Because these O, S and Ni backscattering are difficult to distinguish in first shell as shown in Fig. R7a (Supplementary Fig. 12a), we use wavelet transform to resolve them along k. Fig. R7c (Supplementary Fig. 12c) shows the overview wavelet in the interval $3 \text{ \AA}^{-1} \leq k \leq 11 \text{ \AA}^{-1}$ with Morlet parameters $\eta=10$, $\sigma=1$, and a signal weighting of k^3 . Only one main peak A in Fig. R7c (Supplementary Fig. 12c) is in agreement well with the FT-EXAFS in Fig. R7a (Supplementary Fig. 12a). By optimizing parameters $\eta=1.6$, $\sigma=1$ in the interval $5 \text{ \AA}^{-1} \leq k \leq 8 \text{ \AA}^{-1}$, the resolution along k is improved and the one main peak is resolved to three peaks at $k=5.9$, 6.6 and 7.4 \AA^{-1} , respectively, thereby revealing the presence of Ni-O, Ni-S and Ni-Ni in the first shell (see in Figure R7d, i.e., Supplementary Fig. 12d).”

Fig. R8. The Ni *K*-edge XAS spectra at specific applied voltages of (a) +0.50 and (b) +1.00 V vs. RHE (+0.50 V_P and +1.00 V_P mean +0.50 and +1.00 V in positive sweep of CV) in III, extracted from Fig. 2c, and their linear combination of Ni₃S₂ and NiO. The content of NiO is 13% and 22% at +0.50 and +1.00 V vs. RHE respectively, that is similar to the results obtained by EXAFS fitting as shown in Table R4.

We conducted additional *operando* XAS to ensure NiO still exists during HER conditions. The analysis of *operando* XAS in Fig. 2 support the presence of NiO at positive applied voltages in region III in Figs. 2a, c. This experiment is very important to support the HER mechanism, so we added Fig. R9 as Fig. 3 in the revised manuscript and added new contents in page 10 of the revised manuscript.

Fig. R9. (a) *Operando* XAS at Ni *K*-edge of NiS electrode in chronoamperometry for 30 minutes at 4 consecutively applied potentials of 0.00 V, -1.00 V, 0.50 V, and back in -1.00 V. (b) *Operando* Raman spectra of NiS electrode in chronoamperometry for 30 minutes at a constant applied potential of -0.30 V. (All potentials are vs. RHE).

Comment 9. *The comparison between the experimental and fitted EXAFS spectra should be provided in separate figures for the results reported in Tables S3 and S4 in the Supplemental Materials as well as in Fig. 3(b).*

Response 9 –Thanks for this suggestion. The experimental and fitted EXAFS spectra have been provided in Fig. R10, 11 for the results reported in Supplementary Tables 3, 4. We added Fig. R10, 11 to Supplementary Figs. 13, 14 in page 18 of the revised Supplementary Information. We also moved Fig. 3b in previous manuscript to Supplementary Fig. 12a in revised Supplementary Information with the comparison between experimental and fitted EXAFS spectra. The agreement between the measurements and the fits is good, as confirmed by the low R-factor.

Fig. R10. The *operando* FT-EXAFS and their fitting of NiS at different applied voltages ((a) +1.00, (b) +0.50, (c) 0.00, (d) -0.50, (e) -1.00 and (f) -1.50 V vs. RHE) during the stage of *negative sweep of CV*. The fitting results are shown in Table R1 (Supplementary Table 3).

Fig. R11. The *operando* FT-EXAFS and their fitting of NiS at different applied voltages ((a) -1.50, (b) -1.00, (c) -0.50, (d) 0.00, (e) +0.50 and (f) +1.00 V vs. RHE) during the stage of *positive sweep of CV*. The fitting results are shown in Table R2 (Supplementary Table 4).

Comment 10. *The values of coordination numbers (CNs) should contain errors. The meaning of asterisks should be explained in the captions of Tables. The small values of the Ni-O CNs do not convince for the occurrence of the NiO phase.*

Response 10 – Thanks for the observation. As described in response 6, we have removed the NiO spectral contribution to the fit of spectra in region I and II in Fig. 2c, and only included NiO in region III. The refitted results have been presented in response 5, 6. Beside this, the errors have been introduced in all the CNs used as free parameters of the fit. The asterisks mean that parameter has been kept fixed during the fitting process. This is the case for the Ni-Ni distance.

We have introduced the following statement in Supplementary Note 5 in page 9 of the revised Supplementary Information: “Considering that most of the changes occurs in the first shell, the CN of Ni-S have been used as a free parameter, while for the Ni-Ni we used a fixed CN according to the crystallographic data. The use of a free coordination number for the Ni-Ni gave a very high correlation with the coordination number of Ni-S and made the estimation impossible because the two contributions (Ni-S and Ni-Ni) are enclosed in a single shell and cannot be separated with a limited k-range”

Comment 11. *It looks like Fig. 3(d) contains wrong labels for the spectra: the blue line, probably, corresponds to NiO. One can recommend using the same colors in Fig. 3(b) and (d).*

Response 11 – Thanks for the careful observation and very good suggestions. Figures have been updated as suggested. In the revised manuscript, this Fig. 3d has been moved to page 17 of the Supplementary Information and now it is labelled as Supplementary Fig. 12b.

Comment 12. *The Raman data (Fig. 2e) are available in a short-range, so only two bands around 320 and 480 cm^{-1} are visible in the catalyst. Their assignment cannot be unique. Also, the band at 560 cm^{-1} discussed in the text and attributed to NiOx phase is not present in the Raman spectra. Therefore, the Raman data should be re-analysed. Comparison with the literature data should be done.*

Response 12 – Thanks for your helpful suggestions and we are sorry the confusion induced. We have updated the Fig. 2e (Fig. R12) and extended the range to 1200 cm^{-1} .

Fig. R12. *Operando* Raman spectroscopy of NiS at different applied voltages.

The Raman spectrum (top, black line) at OCP potential in Fig. R12 (Fig. 2e) can be assigned to pristine NiS, with characteristic band appearing at Raman shifts 281 cm^{-1} and 473 cm^{-1} . When HER is performed at negative potential, the phase transition from NiS to NiO/ Ni_3S_2 occurs. As shown in Fig. R12 (Fig. 2e), as the sample was polarized to negative potentials, the characteristic peaks assigned to NiS disappeared, while the characteristic peaks assigned to Ni_3S_2 appear at 221, 303, 322, and 349 cm^{-1} (marked as a, b, c and d, respectively), which indicate the full phase transition from NiS to Ni_3S_2 ⁴. In addition, the new emerged peaks e, f, g, h and i located at 453, 493, 600, 800 and 1060 cm^{-1} are assigned to Ni-O⁵⁻⁸, indicating the formation of NiO. We have added the highlighted text in page 9 of the revised manuscript.

References:

4. Cheng, Z., Abernathy, H. & Liu, M. Raman spectroscopy of nickel sulfide Ni_3S_2 . *J. Phys. Chem. C* **111**, 17997-18000 (2007).
5. Fan, L. *et al.* Molecular functionalization of NiO nanocatalyst for enhanced water oxidation by electronic structure engineering. *ChemSusChem* **13**, 5901-5909 (2020).
6. Bala, N., Singh, H. K., Verma, S. & Rath, S. Magnetic-order induced effects in nanocrystalline NiO probed by Raman spectroscopy. *Phys. Rev. B* **102**, 024423 (2020).
7. Sunny, A. & Balasubramanian, K. Raman spectral probe on size-dependent surface optical phonon modes and magnon properties of NiO nanoparticles. *J. Phys. Chem. C* **124**, 12636-12644 (2020).
8. Radinger, H. *et al.* Importance of nickel oxide lattice defects for efficient oxygen evolution reaction. *Chem. Mater.* **33**, 8259-8266 (2021).

Comment 13. *It is difficult to judge the accuracy of DFT calculations since no comparison with any reference data is provided. It is not clear how well the structural and electronic properties are reproduced, in particular, taking into account that the PBE functional is not hybrid. The position of the Fermi level should be given in Fig. 5(d).*

Response 13 – Thanks for your helpful suggestions. The accuracy of our DFT calculation for the structural and electronic properties of the materials were actually justified by comparing with results from literature. We are sorry we didn't provide detailed discussions. In the revised manuscript, we provide the following descriptions for justification of the accuracy of DFT calculations in Supplementary Note 6 in page 9-11 of the revised Supplementary Information. We also added the Fermi level (set to zero) in Fig. 5e in revised manuscript, *i.e.*, Fig. 5d in previous manuscript.

The crystal structures and electronic structures of NiS and Ni₃S₂ have been previously reported by experiments and theory⁹⁻¹³. Therefore, the crystal structure, lattice parameters, atomic bond distances and angles of NiS and Ni₃S₂ are optimized according to the literature and compared to the results of previous works^{9,10}, as listed Table R5. The good agreement between the optimized lattice parameters and the experimental results suggests our computational methodology is reliable to describe the crystal structure of NiS and Ni₃S₂.

Table R5. The crystal parameters, internal atomic distances, and angles of NiS and Ni₃S₂.

		Computational results	Experimental results ^{9,10}
NiS (P63/mmc)	(a, b, c)	(3.45, 3.45, 5.19)	(3.44, 3.44, 5.35)
	(α , β , γ)	(90.0, 90.0, 120.0)	(90.0, 90.0, 120.0)
	Ni-S	2.38	2.39
	Ni-Ni	2.60	2.68
	< S-Ni-S	(87.0, 93.0, 180.0)	(88.2, 91.8, 180.0)
	< Ni-Ni-Ni	(60.0, 90.0, 120.0, 180.0)	(60.0, 90.0, 120.0, 180.0)
Ni ₃ S ₂ (R32H)	(a, b, c)	(4.08, 4.08, 4.08)	(4.06, 4.06, 4.06)
	(α , β , γ)	(89.4, 89.4, 89.4)	(89.6, 89.6, 89.6)
	Ni-S	(2.26, 2.28)	(2.25, 2.29)
	Ni-Ni	2.54	2.53
	< S-Ni-S	100.6, 103.4, 127.8	(100.8, 127.3)
	< Ni-Ni-Ni	60.0	(60.0, 99.0, 108.1, 148.7)

For the electronic structure, previous experimental and theoretical studies reported that both NiS and Ni₃S₂ have metallic states¹¹⁻¹³. Our DFT calculated electronic structures for NiS (Fig. R13a) and Ni₃S₂ (Fig. R13b) also indicate the metallic state for both materials. Furthermore, the calculated density of state (DOS) also indicates that there are substantial Ni 3d state contributing to the DOS around the Fermi level, also in good agreement with the literatures¹¹⁻¹³. Therefore, our calculation methods can reproduce well the structural and electronic properties of NiS and Ni₃S₂.

Fig. R13. Calculated total and partial density of states of samples: (a) NiS and (b) Ni₃S₂.

We also calculated the Gibbs free energies of H adsorption on pristine Ni₃S₂ and compared the data with other published work¹⁴, which was performed by using PBE functional. Our results are well consistent with the reference data¹⁴. The usage of this methods (GGA-PBE) in our work show very good performance in understanding the reaction mechanism and activity trends observed in experiments. Therefore, the accuracy of DFT calculations used in our work is reliable.

References:

- McWhan, D. B., Marezio, M., Remeika, J. P. & Dernier, P. D. Pressure-temperature phase diagram and crystal structure of NiS. *Phys. Rev. B* **5**, 2552-2555 (1972).
- Vershinin, A. D., Selivanov, E. N., Gulyaeva, R. I. & Sel'menskikh, N. I. Thermal Expansion of Ni₃S₂ in Ni₃S₂-Ni Alloys. *Inorg. Mater.* **41**, 882-887 (2005).
- Wang, J.-H., Cheng, Z., Bredas, J.-L. & Liu, M. Electronic and vibrational properties of nickel sulfides from first principles. *J. Chem. Phys.* **127**, 214705/214701-214705/214708 (2007).
- Krishnakumar, S. R., Shanthi, N. & Sarma, D. D. Electronic structure of millerite NiS. *Phys. Rev. B* **66**, 115105 (2002).
- Lu, Z. W., Klein, B. M. & Singh, D. J. Electronic structure of heazlewoodite Ni₃S₂. *Phys. Rev. B: Condens. Matter* **54**, 13542-13545 (1996).
- Feng, L.-L. *et al.* High-Index faceted Ni₃S₂ nanosheet arrays as highly active and ultrastable electrocatalysts for water splitting. *J. Am. Chem. Soc.* **137**, 14023-14026 (2015).

Referee: 3

Comments to the Author:

Report on "In situ formed NiO_x/Ni₃S₂ Nano interfaces Boost High Hydrogen Evolution Reaction Activity in Alkaline Conditions" by X. Ding, et. al

The authors report the results of a comprehensive experimental and theoretical study exploring the structural transformations that occurring within NiS and which accelerate the hydrogen evolution reaction (HER) under alkaline conditions. This is an important area of work with much interest in the formation of H₂ as an energy storage medium through the electrolysis of water using renewable energy sources. The results presented by the authors suggest that NiS undergoes a phase transformation during the reaction to form an NiO_x/Ni₃S₂ phase, which then acts as a catalyst for the rate-limited step in the HER under alkaline conditions. This is an important observation because understanding the structure on which a reaction occurs is key for the design of improved catalysts. As such, this manuscript will be of broad interest and importance. I recommend publication after the following points have been adequately addressed. Note that given that my expertise lies in theoretical chemistry,

I have only provided comments related to the calculations report by the authors.

Response – Thanks to referee 3 for his/her positive feedback on our work. We are appreciated for the helpful suggestions/comments for improving our manuscript. Followings are our responses to comments.

[The comments are shown in *italic*; responses are in black; all revisions in manuscript and supplementary information are highlighted in dark green.]

Comment 1. *The theoretical methods section in the supporting information do not adequately describe key parts of the calculations. In particular, the authors should describe what kinds of pseudopotentials were used (I assume PAWs, but it isn't stated), what planewave cutoff was used for the density and/or augmentation charges, and the k-point grid used in the calculation. In addition, the authors should outline how zero-point energies and entropies were obtained. I assume through phonon calculations, but it is not stated. It is also important to state whether such calculations (if they were phonon calculations) were performed for the whole system or just in the region near the reactive site.*

Response 1 – Thanks for this suggestion and we are sorry for lack of the information for the calculation methods. We have added methods for the calculations in Supplementary Method 7 in page 4, 5 of the revised Supplementary Information, as follows.

“The generalized gradient approximation (GGA) with the Perdew-Burke-Ernzerhof (PBE) exchange-correlation functional¹⁵ and a 450-eV cutoff for plane-wave basis set were employed to perform all the density functional theory (DFT) computations of the materials within the frame of Vienna *ab initio* simulation package (VASP)^{16,17}. The projector-augmented plane wave (PAW) was used to describe the electron-ion interactions^{18,19}. 5×5×1 Monkhorst-Pack grid k-point were employed for geometric optimization of the slab surface. The convergence threshold was set to 10⁻⁵ eV and 0.02 eV/Å for energy and force, respectively. We opened the spin-polarization during the calculations to obtain accurate results of the total energy and density of states (DOS). The (110) surface of NiS and ($\bar{2}10$) surface of Ni₃S₂ with ~15 Å vacuum were simulated as the catalytic interface. Van der Waals interactions were considered using the Grimme’s D3-type of the semiempirical method because of long-range interaction present between the system and gas molecules.”

“Based on computational hydrogen electrode (CHE) model²⁰, the Gibbs free energies were calculated by

$$\Delta G = \Delta E + \Delta E_{\text{ZPE}} - T\Delta S$$

where ΔE is the adsorption energy of the adsorbed intermediates, and ΔE_{ZPE} and ΔS are the difference in the zero-point energy and the change in entropy before and after adsorption of various intermediates, respectively. The zero-point energies and entropies of the reaction species were calculated from the vibrational frequencies. For gas phase molecule, the entropy term can be expressed as the sum of the translational, rotational, and vibrational contributions, whereas for adsorbates the translational and rotational entropy were not taken into account due to negligible contributions. Specifically, the vibration frequency of hydrogen adsorption is 3211.45 cm⁻¹, which is not sensitive to the adsorption sites. The entropy is obtained based on the equation:

$$S(T) = \sum_{i=1}^{3N} \left[-R \ln \left(1 - e^{-\frac{h\nu_i}{k_B T}} \right) + \frac{N_A h \nu_i}{T} \frac{e^{-h\nu_i/k_B T}}{1 - e^{-h\nu_i/k_B T}} \right]$$

where R stands for the universal gas constant, k_B is the Boltzmann constant, h is Planck's constant, N_A is Avogadro's number, ν_i represents the frequency and N is the number of adsorbed atoms. In addition, during these frequency computations, all atoms of substrate were rigidly constrained so that no additional degrees of freedom from catalysts are introduced into the reaction system. Therefore, such calculations were performed just for the reaction intermediates, and the contributions from the catalysts to ΔE_{ZPE} and ΔS are neglected.”

Reference:

15. Perdew, J. P., Burke, K. & Ernzerhof, M. Generalized gradient approximation made simple. *Phys. Rev. Lett.* **77**, 3865-3868 (1996).
16. Kresse, G. & Hafner, J. Ab initio molecular dynamics for liquid metals. *Phys. Rev. B* **47**, 558-561 (1993).
17. Kresse, G. & Hafner, J. Ab initio molecular-dynamics simulation of the liquid-metal-amorphous-semiconductor transition in germanium. *Phys. Rev. B* **49**, 14251-14269 (1994).
18. Blöchl, P. E. Projector augmented-wave method. *Phys. Rev. B* **50**, 17953-17979 (1994).
19. Kresse, G. & Joubert, D. From ultrasoft pseudopotentials to the projector augmented-wave method. *Phys. Rev. B* **59**, 1758-1775 (1999).
20. Man, I. C. *et al.* Universality in oxygen evolution electrocatalysis on oxide surfaces. *ChemCatChem* **3**, 1159-1165 (2011).

Comment 2. *In the first reaction step in Figure 3a, the authors show S leaching from the material and indicate that oxygen should fill the vacancy. However, the middle image in that figure states that the material is Ni₃S₂ and shows no oxygen. So, how does oxygen factor in? I found this pretty confusing.*

Response 2 – Thank you for this helpful comment and sorry for the confusion. In the revised manuscript, we have changed Fig. 3a to Fig. 2f, and made some modifications.

We went back to synchrotron and re-performed the *operando* XAS experiment. Details of the new results are provided in Fig. R14a-b and Supplementary Table 3, 4. We found substantial oxidation of Ni occur at applied voltage of above +0.27 V *vs.* RHE during positive sweep of CV. Therefore, there is only S leaching in the first reaction step, and oxygen filling mainly occurs during positive sweep. Therefore, we revised the schematic illustration for the phase transition shown in Fig. R14c (Fig. 2f), and added the following description in page 9, 10 in the revised manuscript.

Fig. R14. (a) *Operando* Ni *K*-edge XAS for NiS catalyst as a function of applied voltages during a cycle of CV. Top panel shows a 3-dimensional view of Ni *K*-edge XAS spectra; bottom shows a 2-dimensional projection from 3-dimensional view. The oxidation of Ni only occurs in the region III when the applied voltage is above +0.27 V. (b) The coordination number of Ni as a function of applied voltages during a cycle of CV. The results show the coordination of Ni by O only occurs in the region III. Combining (a) and (b), the filling of oxygen doesn't happen in the first step. (c) Schematic illustration for the phase transition of NiS catalyst during HER measurement.

(step 1) the electrochemical reduction of NiS to Ni₃S₂ at -0.27 V vs. RHE, which is driven by the thermodynamic instability of NiS under HER conditions; (step 2) under more reductive HER conditions, S leaching leads to the formation of S vacancies in Ni₃S_{2-δ}; (step 3) as the sample is polarized to more positive potentials, substantial oxidation occurs at +0.27 V vs. RHE, and S vacancies are occupied by the oxygen species from the media (water/OH⁻ ions), leading to the formation of an intimately mixed phase of NiO/Ni₃S₂.

Comment 3. In Figure 5c, the authors show the structure of NiOx/Ni₃S₂ and suggest a mechanism. The mechanism relies on water binding through the oxygen atom to a nickel

atom that is bonded to an oxygen and a sulfur. Based on the structure provided, it's not clear that a site facilitate this process is present and the images provided in the supporting information (Figures S9 - S11) don't really support this mechanism, either. It would be helpful if the authors indicated more accurately in the main text how this reaction occurs. In addition, the authors should provide the coordinates of all calculated structure to the supporting information so others can easily reproduce the results.

Response 3 – Many thanks to the referee for this very helpful suggestion. Our results suggest the interfacial Ni site (O-Ni-S) of NiO/Ni₃S₂ facilitate the water dissociation and nearby S sites of Ni₃S₂ facilitate hydrogen generation. We revised Fig. 5 in the revised manuscript and added Supplementary Figs. 18-20 in page 24-26 of the revised Supplementary Information to illustrate more clearly the reaction mechanism and energetics. We also added the following discussions on the reaction mechanism and energetics in page 12, 13 of the revised manuscript. In addition, we provided the coordinates of all calculated structure to the Supplementary Data file/document.

Firstly, as shown in Fig. R15a (Fig. 5a), we propose a mechanism of NiO/Ni₃S₂ for HER: we propose the catalytic mechanism for alkaline HER over NiO/Ni₃S₂ as schematically shown in Figure R15a (Fig. 5a in revised manuscript). The water molecule firstly adsorbs at the interfacial Ni sites (O-Ni-S) (step I), then dissociates into adsorbed OH* and H* (steps II and III). Subsequently, another proton from adjacent H₂O molecule will react with the adsorbed H* to generate H₂ (steps IV-VI).

Fig. R15. (a) The schematic diagram for HER mechanism of NiO/Ni₃S₂. (b) The calculated adsorption energies of H₂O at different sites of NiO/Ni₃S₂. One site is the interfacial Ni site that is bonded to an oxygen and a sulfur atom (defined as O-Ni-S), the other is Ni site that is bonded to two oxygen atoms (defined as O-Ni-O).

Secondly, DFT calculation were performed to further verify the reaction energetics. Detailed calculation methods and structural optimization are provided in Supplementary Method 7 and Supplementary Figs. 18-20. (1) Step I: It is well known that in alkaline HER, the dissociation of water molecules is the sluggish step^{21,22}. Hence, we first consider adsorption energies of H₂O molecules at different atomic sites of NiO/Ni₃S₂ surface. As shown in Fig. 5b (Fig. R15b), the H₂O molecules has a larger adsorption energy of 0.93 eV at the interfacial Ni site which is bonded to an oxygen and a sulphur (denoted as O-Ni-S) than that (0.72 eV) at the Ni site which is bonded to two oxygen atoms (denoted as O-Ni-O), indicating a favourable adsorption of H₂O molecule at the interfacial Ni site (O-Ni-S).

Fig. R16. The calculated free energy diagrams of H₂O molecule dissociation and their corresponding atomic configurations on different sites of NiO/Ni₃S₂. One site is the interfacial Ni site that is bonded to an oxygen and a sulfur atom (defined as O-Ni-S), the other is Ni site that is bonded to two oxygen atoms (defined as O-Ni-O).

(2) Step II and III: Fig. 5c (Fig. R16 *i.e.*, Supplementary Fig. 18) shows the free energy diagrams for H₂O dissociation into OH* and H* at the interfacial Ni site (O-Ni-S) and O-Ni-O site. The interfacial Ni site (O-Ni-S) exhibits a low energy barrier of 1.05 eV for the water dissociation, while the energy barrier is 2.70 eV for the O-Ni-O site. This indicates

the interfacial Ni (O-Ni-S) provides optimal site for water dissociation, leading to a faster proton supply for the following H₂ generation.

Fig. R17. The calculated free energy diagram of H adsorption and their corresponding atomic configurations on S sites, Ni sites and O sites of NiO/Ni₃S₂. We also calculated the free energy of H adsorption on other sites as show in Supplementary Fig. 20.

(3) Step IV, V and VI: Furthermore, Fig. 5d (Fig. R17, *i.e.*, Supplementary Fig. 19) shows the calculated ΔG_{H^*} at different sites of NiO/Ni₃S₂, and their corresponding atomic configurations of H adsorption are shown in supplementary Fig. 19. The calculated results reveal that the ΔG_{H^*} value at the S site (0.08 eV) is closer to the thermal neutral position (zero) in comparison with that at the Ni site (0.27 eV) and O site (-1.21 eV), implying an optimal H* adsorption and desorption at S site, agreeing well with S site as active site for high HER activity.

Based on above discussions, it is clear that the interfacial Ni sites (O-Ni-S) of NiO/Ni₃S₂ facilitate the water dissociation and nearby S sites of Ni₃S₂ facilitate hydrogen generation. We have illustrated this process and reaction energetics in Fig. 5a-d in the revised

manuscript and Supplementary Figs. 18-20 in the Supplementary Information. We also added detailed discussion in pages 12, 13 in the revised manuscript.

Reference:

21. Hu, C., Zhang, L. & Gong, J. Recent progress made in the mechanism comprehension and design of electrocatalysts for alkaline water splitting. *Energy Environ. Sci.* **12**, 2620-2645 (2019).
22. Mahmood, N. *et al.* Electrocatalysts for Hydrogen Evolution in Alkaline Electrolytes: Mechanisms, Challenges, and Prospective Solutions. *Adv Sci (Weinh)* **5**, 1700464 (2018).

Comment 4. *The mechanism outlined in the bottom of Figure 5c involves a proton transferring to sulfur. It seems that transfer to oxygen would be preferred. Perhaps I missed the explanation of why this is the case. However, it is important to explain why the oxygen atom isn't protonated, since most people would intuitively expect that to be preferred*

Response 4 – Thanks for the valuable comment. We agree with you that proton transfer to oxygen would be preferred, and our calculation results also support this view. In previous Fig. 5, we didn't draw proton on O site, because O site is not an active center for HER, and we just focused on the cyclic process on active site to show HER mechanism. In order to make Fig. 5 more clear, we have updated Fig. 5 in the revised manuscript, and added H* on O site (marked in blue) without process of desorption (see Fig. R18, *i.e.*, Fig. 5a). We also added description about this in page 13 in the revised manuscript.

Fig. R18. The schematic diagram for HER mechanism of NiO/Ni₃S₂. We added the H on the O sites (marked in blue). The H* on O sites don't participate in the desorption of H* for releasing H₂ because of the more negative value of ΔG_{H^*} (-1.21 eV). Therefore, we just draw it on O sites, but no process of desorption.

Fig. 5d (Fig. R19, i.e., Supplementary Fig. 19) shows the calculated ΔG_{H^*} at different sites of NiO/Ni₃S₂, and their corresponding atomic configurations of H* adsorption are shown in Supplementary Fig. 19. The calculated results reveal that the ΔG_{H^*} value at S site (0.08 eV) is close to the thermal neutral position (zero) in comparison with that at Ni site (0.27 eV) and O site (-1.21 eV), implying optimal H* adsorption and desorption at S site, agreeing well with S site as active site for high HER activity.

As shown in Fig. R19 (Fig. 5d in revised manuscript and Supplementary Fig. 19), Our results also indicate that although H* prefer to adsorb on the O site, the desorption of H* from O is difficult because of the too strong adsorption of H* at O site ($\Delta G_{H^*} = -1.21$ eV). Therefore, in Fig. 5a (Fig. R18), we show the adsorption of H* (marked in blue) on O site, but no cyclic process to generate H₂.

Fig. R19. The calculated free energy diagram of H* adsorption and their corresponding atomic configurations on S sites, Ni sites and O sites of NiO/Ni₃S₂. The value of ΔG_{H^*} (-1.21 eV) of H* on O sites is too negative, so the adsorption of O to H* is too strong that H* can't desorb from O for releasing H₂.

Reviewer #2 (Remarks to the Author):

The revised version of the manuscript was significantly improved, however, several important questions are still open.

The detailed description of the EXAFS simulations/fits is fully missing which undermines the credibility of the data obtained.

1) It is not clear what software and how was used for the data analysis. What approximations were employed? Did the authors test the theory used in the fits on any reference samples?

2) The fitting results in Figs. S13 and S14 show only the modulus of the Fourier transforms (FTs) thus the information on the phase is fully missing. The imaginary part of FTs should be shown or, even better, the plots of EXAFS spectra in k-space should be additionally presented.

3) The same is true for the result of the fit reported in Fig. S12. In this case (and in Fig. S14 (e,f)), the authors should also prove that the three-component fit is statistically significant.

4) Finally, the too-short Ni-O distances reported may be due to (1) the harmonic model being used in the case of strong disorder or (2) the "inaccurate" choice of the kinetic energy origin (E_0) for the photoelectron wavenumber k . Please, check the data analysis.

Reviewer #4 (Remarks to the Author):

The work describes the experimental finding of a good HER catalysts under alkaline conditions, which are in situ reconstructed from NiS. As a theoretician, I mainly checked the theoretical part of the work. However, I am not satisfied with the computation results, which are unclear on both the model and the kinetics. This part is not scientifically sound.

1. The catalyst structure. There is very limited experimental evidences on the atomic structure of the catalyst. It appears for quite a certain the catalyst is amorphous. The theoretical part utilizes a rather simple layer-by-layer crystalline model, which is obviously not consistent with experiment. There is even no thermodynamics analysis on the stability of the catalyst structure from DFT.

2. The water dissociation barrier is at least 1.07 eV, which is too high for reaction to occur at the ambient condition.

3. The H adsorption energy on three sites are evaluated, which shows that H on O site can be very stable. Therefore, this is a good indication that the current model is not right. Even the H coverage has not been explicitly explored.

From my point of view, the theoretical part of the work does not add new insights, but create more confusions on the exact catalyst structure and on the reaction mechanism.

RESPONSE TO REVIEWERS' COMMENTS

Reviewer #2 (Remarks to the Author):

The revised version of the manuscript was significantly improved; however, several important questions are still open. The detailed description of the EXAFS simulations/fits is fully missing which undermines the credibility of the data obtained.

Response: Thanks a lot for Reviewer #2's positive comment. According to your kind suggestion, we have further revised our manuscript and provided the detailed description of the EXAFS simulations/fits. Hopefully, the current version can be satisfied.

Q1. *It is not clear what software and how was used for the data analysis. What approximations were employed? Did the authors test the theory used in the fits on any reference samples?*

Response: Thanks a lot for Reviewer #2's questions. The software utilized for EXAFS analysis is ARTEMIS from the Demeter package. The crystallographic information files (CIFs) for NiS and Ni₃S₂ were employed in the FEFF 6 calculation to derive the scattering paths. The software employs the path expansion approximation, treating the EXAFS spectrum as the sum of contributions from scattering geometries involving two or more atoms in a cluster surrounding the absorbing atom (Ravel et al., *Journal of Physics: Conference Series* 2013, **430**, 012006).

Note that the data were normalized through background subtraction using the ATHENA software within the Demeter package before the EXAFS analysis. As shown in Supplementary Note 1, "EXAFS simulations procedure," (Page 23 and 24 in Supplementary Information), the EXAFS data were analyzed in the k-range of 3–11 Å⁻¹, and the Fourier Transform was executed using a Hanning window. The data were fitted in the R-range of 1–2.6 Å using a multiple k-weight fitting. The fits employed the most intense single scattering paths, Ni-S and Ni-Ni, associated with NiS and Ni₃S₂ phases, in accordance with the results obtained from XRD, Raman, and XANES investigations. The amplitude reduction factor S₀² was determined from a reference Ni foil, and its value of 0.8 was fixed for all scattering paths. This procedure is also widely adopted in the analysis of EXAFS data, particularly when experiments are conducted under the same conditions.

Regarding the model for fitting the data, we would like to clarify as following: 1) only the most intense single scattering paths were utilized, 2) the coordination number of the Ni-Ni shell was fixed based on crystallographic data of NiS or Ni₃S₂, while the coordination number of the first shell was treated as a free parameter. This approach was necessary because, when leaving the coordination number of the second shell as a free parameter in the fit, a high correlation was observed with the Ni-S shell. The contributions of Ni-S and Ni-Ni are included in a single shell and cannot be distinguished within a limited k-range. Conversely, R and σ of the first and second shell were treated as independent parameters.

Regarding the analysis of reference samples, a good agreement was achieved for the reference NiS, as confirmed by the values in Supplementary Table 2 (potential 0.00 in region I corresponding to the NiS sample) on Page 13 in Supplementary Information.

We have added the above information and please see them in Supplementary Note 1 on Page 23 and 24 in the Supplementary Information.

Q2. *The fitting results in Figs. S13 and S14 show only the modulus of the Fourier transforms (FTs) thus the information on the phase is fully missing. The imaginary part*

of FTs should be shown or, even better, the plots of EXAFS spectra in k -space should be additionally presented.

Response: Thanks a lot for Reviewer #2's kind suggestion. We have provided the imaginary parts of the Fourier Transforms (FTs) and the extended X-ray Absorption Fine Structure (EXAFS) spectra in k -space. As shown in Fig. R1, a distinct shift in the imaginary part is observed when transforming NiS into Ni₃S₂, which may be attributed to the shorter Ni-S bond distance in Ni₃S₂ along with a reduced coordination number (CN) of the Ni-S. The analysis of the imaginary part clearly indicates a significant transformation of the initial NiS compound at positive potential (Fig. R2).

Fig. R1 (Supplementary Fig. 8 in the revised Supplementary Information) | The *operando* FT-EXAFS, imaginary part of FT-EXAFS, EXAFS in k -space and their fitting of NiS at different applied voltages: (a) +1.00, (b) +0.50, (c) 0.00, (d) -0.50, (e) -1.00 and (f) -1.50 V vs. RHE during the stage of negative sweep of cyclic voltammetry (CV).

Fig. R2 (Supplementary Fig. 9 in the revised Supplementary Information) | The *operando* FT-EXAFS, imaginary part of FT-EXAFS, EXAFS in k -space and their fitting of NiS at different applied voltages: (a) -1.50 , (b) -1.00 , (c) -0.50 , (d) 0.00 , (e) $+0.50$ and (f) $+1.00$ V vs. RHE during the stage of positive sweep of CV.

We have updated the Supplementary Figs. 13 and Fig. 14 and re-named them as Supplementary Fig. 8 and Fig 9 on Page 12 and 14 in the revised Supplementary Information.

Q3. The same is true for the result of the fit reported in Fig. S12. In this case (and in Fig. S14 (e, f)), the authors should also prove that the three-component fit is statistically significant.

Response: Thanks a lot for Reviewer #2's comment. In our previous XANES linear combination fitting (LCF), where NiO served as a reference for the oxidized portion (Fig. R3 and Table R1), we obtained a good model incorporating 13% or 22% of NiO in addition to Ni₃S₂. This model effectively represented the XANES spectra at +0.50 and +1.00 V during the positive sweep of CV, indicating the presence of Ni-O compounds (we assumed to be in the form of NiO).

Fig. R3 (Fig. 2c in the revised manuscript) | The Ni K-edge XAS spectra at specific applied voltages of +1.00 V (1 V_N means +1.00 V in negative sweep of CV), -1.50 V (-1.50 V_N means -1.50 V in negative sweep of CV), and +0.50 V, +1.00 V (+0.50 V_P and +1.00 V_P mean +0.50 V and +1.00 V in positive sweep of CV) in region I, II and III respectively.

Moving on to the EXAFS fit using the three-component model (Ni-O, Ni-S, and Ni-Ni), fixing the weights of the Ni-O shell (13% or 22%) and the Ni-S shell (87% or 78%) based on XANES LCF would result in a statistically significant improvement in the fit. While introducing new variables often leads to an improved fit, we employed the F-test to demonstrate the statistical significance of the improvement. The F-test considers the degrees of freedom in the fit (*i.e.*, the number of independent points minus variables) and the reduced chi-squared of the fit. In our analysis, the three-component model yielded a probability of approximately 90% for the +1.00V sample in Table R2, indicating that it significantly outperformed the two-component model for the data.

Furthermore, it should be noted that when the presence of distinct compounds (10% or more) is evident in XANES spectra, this information should be extractable through the analysis of the EXAFS spectra. This is particularly relevant for compounds with markedly different bond lengths, such as Ni-O and Ni-S.

Table R1 (Supplementary Table 4 in the revised Supplementary Information) | Composition change for NiS during *operando* XAS at +0.50 and +1.00 V vs. RHE.

Applied potential (V vs. RHE)	NiO (%)	Ni ₃ S ₂ (%)	R-factor
+0.50	13±0.3	87±1.7	0.0006716
+1.00	22±0.5	78±1.6	0.0004868

Table R2 (Supplementary Table 3 in the revised Supplementary Information) | *Operando* EXAFS fitting parameters at the Ni-K edge of NiS in different applied voltages during the stage of positive sweep of CV. ($S_0^2 = 0.8$)

Applied voltage (V vs. RHE)	Path	CN	R (Å)	σ^2 (Å ²)	R-factor
-1.50 (region II)	Ni-S	3.3±0.6	2.262±0.013	0.005±0.003	0.003
	Ni-Ni	4.00*	2.50±0.01	0.007±0.001	
-1.00 (region II)	Ni-S	2.9±0.8	2.262±0.016	0.005±0.004	0.005
	Ni-Ni	4.00*	2.502±0.012	0.007±0.001	
-0.50 (region II)	Ni-S	2.9±0.6	2.258±0.012	0.006±0.001	0.002
	Ni-Ni	4.00*	2.495±0.008	0.007±0.001	
0.00 (region II)	Ni-S	3.0±0.6	2.258±0.017	0.006±0.003	0.006
	Ni-Ni	4.00*	2.50±0.01	0.006±0.001	
+0.50 (region III)	Ni-S	4 (x=0.87)	2.267±0.015	0.005±0.001	0.009
	Ni-Ni	4.00*	2.509±0.015	0.010±0.002	
	Ni-O	6 (1-x=0.13)	2.03±0.05	0.005±0.001	
+1.00 (region III)	Ni-S	4 (x=0.78)	2.286±0.014	0.005±0.002	0.011
	Ni-Ni	4.00*	2.51±0.002	0.013±0.002	
	Ni-O	6(1-x=0.22)	2.00±0.065	0.005±0.002	

CN: coordination number. **R:** bond distance. **σ^2 :** Debye-Waller factors. **R factor:** goodness of fit. The asterisks (*) mean that parameter is fixed during the fitting process. The 'x' and '1-x' represent the weight of Ni-S path and Ni-O path, respectively.

Q4. Finally, the too-short Ni-O distances reported may be due to (1) the harmonic model being used in the case of strong disorder or (2) the “inaccurate” choice of the kinetic energy origin (E_0) for the photoelectron wavenumber k . Please, check the data analysis.

Response: Thanks a lot for Reviewer #2’s comments. We would like to clarify that both the EXAFS signals and the Morlet transform are depicted on a phase-uncorrected scale. Consequently, the shells appear at length values considerably shorter than the accurate values obtained from the fit, which undergo phase correction. The Ni-O bond distances derived from the fit are measured at 2.03 ± 0.05 Å and 2.00 ± 0.07 Å (Table R2). These values agree very well with typical Ni-O bond distances (Anspoks et al., *Solid State Commun.* 2010, **150**, 2270; Singh et al., *Solid State Commun.* 2017, **259**, 40; Zhao et al., *Materials Advances* 2021, **2**, 4667).

Reviewer #4 (Remarks to the Author):

The work describes the experimental finding of a good HER catalysts under alkaline conditions, which are in situ reconstructed from NiS. As a theoretician, I mainly checked the theoretical part of the work. However, I am not satisfied with the computation results, which are unclear on both the model and the kinetics. This part is not scientifically sound.

Response: Thanks a lot for Reviewer #4’s comments. According to your suggestion, we have re-constructed a more realistic model and conducted DFT calculations to reveal more insight in our system. Accordingly, we have thoroughly revised our manuscript. We believe that the revised version could be satisfied.

Q1. *The catalyst structure. There is very limited experimental evidence on the atomic structure of the catalyst. It appears for quite a certain the catalyst is amorphous. The theoretical part utilizes a rather simple layer-by-layer crystalline model, which is obviously not consistent with experiment. There is even no thermodynamics analysis on the stability of the catalyst structure from DFT.*

Response: We appreciate Reviewer #4’s insightful question. According to the characteristic pattern of XRD, the Ni_3S_2 is determined to be crystalline. While for NiO, the combination of XRD, XPS, XAS and Raman results are hard to determine whether the NiO is crystalline or amorphous. Nevertheless, according to your suggestion, we can roughly determine the stability of NiO *via* a thermodynamics analysis. We have performed *ab initio* molecular dynamics (AIMD) to calculate the most likely stable structure of NiO. AIMD calculation of supercell of NiO ($4 \times 4 \times 4$) under the temperature of 3000 K (melting points 2260.15 K) govern by Nose-Hoover thermostat for 50 ps was performed to obtain an amorphous structure (Fig. R4a) (Zhao et al., *Phys. Rev. B* 2005, **71**, 085107; Lee et al., *Phys. Rev. Lett.* 2011, **107**, 145702). Afterward, the amorphous structure was optimized under system temperature from 1000 K to 3000 K for 50 ps to simulate annealing process. Once the temperature was decreased from 1000 K to 300 K (room temperature), it can be found that the amorphous structure would finally convert into face-centered cubic (FCC) structure (Fig. R4b) (Qiao et al., *J. Am. Chem. Soc.* 2018, **140**, 12256; Branicio et al., *Phys. Rev. Mater.* 2018, **2**, 043401). Therefore, the employed crystalline NiO model would be reasonable. Note that the final NiO model *via* AIMD and DFT simulation underwent a geometrical optimizations process in terms of lattice parameters and ions position.

We have added the above discussion and please see the Note below Supplementary Fig. 15 on page 21–22 in the revised Supplementary Information.

Fig. R4 (Supplementary Fig. 15 in the revised Supplementary Information) | (a) AIMD Nose-Hoover under 3000 K for 50 ps to obtain an amorphous structure of NiO. **(b)** AIMD Nose-Hoover from 1000 K to 300 K to simulation annealing with NVT ensemble for 50 ps. The result indicates the stable state of NiO is crystalline.

Q2. *The water dissociation barrier is at least 1.07 eV, which is too high for reaction to occur at the ambient condition.*

Response: We greatly appreciate Reviewer #4's constructive comment. In the revised manuscript, we have re-built the heterostructure of Ni₃S₂ and NiO with O sites covered by H (marked as Ni₃S₂/NiO) as a new model and re-calculated the energy barrier. As shown in Fig. R5, the energy barrier for water dissociation at the interfacial Ni-S sites of Ni₃S₂/NiO is 0.11 eV, which is suitable for the occurrence of reaction at the ambient condition. Note that energy barrier of 0.11 eV is much lower than those for water dissociation on either individual Ni₃S₂ (0.42 eV) or individual NiO (0.51 eV), suggesting that the facile water dissociation on the mixed phase of Ni₃S₂/NiO.

Fig. R5 (Fig. 5a in the revised manuscript) | Gibbs free energy diagrams of alkaline HER pathway at Ni-Ni sites of NiO, Ni-S sites of Ni₃S₂ and the interfacial Ni-S sites of Ni₃S₂/NiO.

We have also added the above discussion and please see it on Pages 13–15 in the revised manuscript.

Q3. *The H adsorption energy on three sites are evaluated, which shows that H on O site can be very stable. Therefore, this is a good indication that the current model is not right. Even the H coverage has not been explicitly explored. From my point of view, the theoretical part of the work does not add new insights, but create more confusions on the exact catalyst structure and on the reaction mechanism.*

Response: Thanks a lot for Reviewer #4’s critical comment. According to your suggestion, we have employed a heterostructure of Ni₃S₂ and NiO with O sites covered by H as a new model (marked as Ni₃S₂/NiO) for DFT calculations (Fig. R6).

Fig. R6 (Supplementary Fig. 14 in the revised Supplementary Information) | The optimized structure models for theoretical calculations. (a) Ni₃S₂ (marked as Ni₃S₂); (b) NiO with the O sites covered by H (marked as NiO); (c) Ni heterostructure slab with the O sites covered by H (marked as Ni₃S₂/NiO).

The new Ni₃S₂/NiO model with O sites covered by H can be constructed due to the fact that the adsorption of H on surface unsaturated oxygen sites for both Ni₃S₂/NiO and NiO is spontaneous and too strong, as revealed by the calculated values of ΔG_{H^*} in Fig. R7.

Fig. R7 (Supplementary Fig. 16 in the revised Supplementary Information) | The calculated Gibbs free energy diagrams of H* adsorption at O sites of Ni₃S₂/NiO and NiO. The results indicate the adsorption of H on surface unsaturated oxygen sites for both Ni₃S₂/NiO and NiO is spontaneous and too strong.

Based on this new model, we have also calculated the energy barrier of water dissociation and the adsorption energy of H for pure Ni₃S₂, heterostructured Ni₃S₂/NiO and pure NiO catalysts. As shown in Fig. R8, the energy barrier for water dissociation at the interfacial Ni-S sites of Ni₃S₂/NiO is 0.11 eV, which is much lower than those for water dissociation at single Ni₃S₂ (0.42 eV) and single NiO (0.51 eV), suggesting that the formed heterostructure is favorable to the dissociation H₂O molecule. In addition, the adsorption energy of H* on interfacial S sites for Ni₃S₂/NiO is -0.18 eV, which is closer to the optimal value (0 eV) in comparison with those values (-0.31 eV for Ni₃S₂ and 0.39 eV for NiO). These theoretical calculation results further verified that the formed heterostructure can indeed enhance the catalytic activity during HER process, which is consistent with the electrochemical measurements.

We have added the above discussion and please see it on Pages 13–15 in the revised manuscript.

Fig. R8. (Fig. 5a in the revised manuscript) | Gibbs free energy diagrams of alkaline HER pathway at Ni-Ni sites of NiO, Ni-S sites of Ni₃S₂ and the interfacial Ni-S sites of Ni₃S₂/NiO.

Reviewer #2 (Remarks to the Author):

The revised version of the manuscript was improved addressing all questions. Therefore, I can recommend its acceptance for publication.

Reviewer#5

The manuscript presented in-situ formed Ni₃S₂/NiO interfaces during HER process in alkaline conditions. The process and relative mechanism of the transformation of NiS to Ni₃S₂/NiO were discussed clearly. However, in the present form the manuscript is not suitable for publication in Nature Communications. Some important issues are suggested to be addressed carefully.

1). Based on the operando XAS spectroscopy results, the formation of NiO occurred at positive potential (+0.27 V vs. RHE), and its amount of the NiO was increased with an increased the positive potential. Some issues were caused by the experimental results.

First, at the positive potential, the HER current density was too small, and so the transformation did not occur in the HER potential region.

Second, based on the phase transformation the catalysts could be considered to be pre-catalysts. However, as the experimental results shown (Fig.3a and b), "When the sample is brought back to HER conditions at -1.00 V, the content of NiO seems to decrease but still exist (the inset in Fig. 3a) as the spectrum can be fitted with a linear combination of 94% of Ni₃S₂ spectral features and 6% of NiO spectral features after 30 minutes under the reaction". Therefore, when Ni₃S₂/NiO interfaces were applied for HER at negative potentials, the formed NiO would be disappeared especially at operation of HER for a long time.

Third, the main findings and discussion in the work were based on the transformation of Ni₃S₂/NiO interfaces, including the formation process and the enhanced HER mechanism of the interfaces. However, at a large HER current density, the content of NiO was gradually decreased and possibly disappeared for a long-term operation. Therefore, the enhanced mechanism may be different from that proposed in this work when the HER operated for a long-term period (such as the increased content of the S vacancies).

Fourth, "Additionally, the catalytic activity of the NiS-40 sample is very stable, showing no overpotential increase over 25 hours of continuous operation at a current density of 200 mA cm⁻² in 1 M KOH (Supplementary Fig. 5c)." Based on the comment mentioned above, the structure of the HER-post catalysts is suggested to be characterized.

2). Compared to the most reported catalyst, the HER activity of the Ni₃S₂/NiO interfaces (10 mA cm⁻²@95 mV) in this work had no obvious advantages. In fact, the overpotentials of most reported catalysts in the alkaline conditions were lower than that of Pt/C.

3) Based on the HRTEM image (Fig. 1d), the identification of NiO region was lack of evidence.

RESPONSE TO REVIEWERS' COMMENTS

Reviewer #2 (Remarks to the Author):

The revised version of the manuscript was improved addressing all questions. Therefore, I can recommend its acceptance for publication.

Response: We greatly appreciate Reviewer #2 for the positive feedback and recommendation for publication.

Reviewer#5 (Remarks to the Author):

The manuscript presented in-situ formed Ni₃S₂/NiO interfaces during HER process in alkaline conditions. The process and relative mechanism of the transformation of NiS to Ni₃S₂/NiO were discussed clearly. However, in the present form the manuscript is not suitable for publication in Nature Communications. Some important issues are suggested to be addressed carefully.

Response: Thanks a lot for Reviewer #5's very helpful comments. According to your suggestion, we have further revised manuscript and provided the longer *operando* experiment as well as the characterization of HER-post catalyst. Hopefully, the current version can be satisfied.

Q1. *Based on the operando XAS spectroscopy results, the formation of NiO occurred at positive potential (+0.27 V vs. RHE), and its amount of the NiO was increased with an increased the positive potential. Some issues were caused by the experimental results.*

Response: We appreciate Reviewer #5's insightful question. Although the formation of NiO occurred at positive potential (+0.27 V vs. RHE) based on the *operando* XAS spectroscopy results (Fig. 2a), the formed NiO remains present at negative potential where the HER can take place. We also performed long-time *operando* Raman experiments, and the results provide further evidence of the existence of NiO at negative potential. We believe the new experimental investigations and the revised manuscript could clarify the concerning raised by Reviewer #5.

Q1.1. *First, at the positive potential, the HER current density was too small, and so the transformation did not occur in the HER potential region.*

Response: Thanks a lot for Reviewer #5's comments. The formation of NiO at positive potential may be responsible for the increased catalytic activity of the electrode. We have performed additional *operando* Raman experiment at different applied potential to confirm the occurrence of the transformation in the HER potential region. As depicted in Fig. R1, a notable presence of NiO is observed at an operating potential of 0.4 V vs. RHE, consistent with the results of *operando* XAS presented in Fig. 2a of the main text. Followed by operating at 0.1, 0, -0.1, -0.2 and -0.3 V vs. RHE, respectively, the NiO is reduced, but its content quickly levels off at a certain magnitude. Within these applied potential regions, the HER current density varies from negligible level to several hundred milliamperes per square centimeter. This observation suggests that a transformation from NiS to Ni₃S₂/NiO would occur within the HER potential region.

Fig. R1 | The *operando* Raman spectra were acquired at different operating potential of 0.4, 0.1, 0.0, -0.1, -0.2 and -0.3 V vs. RHE.

Q1.2. *Second, based on the phase transformation the catalysts could be considered to be pre-catalysts. However, as the experimental results shown (Fig.3a and b), “When the sample is brought back to HER conditions at -1.00 V, the content of NiO seems to decrease but still exist (the inset in Fig. 3a) as the spectrum can be fitted with a linear combination of 94% of Ni₃S₂ spectral features and 6% of NiO spectral features after 30 minutes under the reaction”. Therefore, when Ni₃S₂/NiO interfaces were applied for HER at negative potentials, the formed NiO would be disappeared especially at operation of HER for a long time.*

Response: Thanks a lot for Reviewer #5’s insightful comments. The amount of bulk NiO may decrease, but it will not disappear. To confirm the existence of NiO, we have conducted long-time *operando* Raman spectra measurements. Due to the excessive generation of bubbles at -1 V vs. RHE, which significantly impedes the acquisition of reliable *Raman* spectra, we opted to collect the Raman spectra at a constant applied potential of -0.3 V vs. RHE. under HER condition. The electrodes already show a high HER current density of above 300 mA cm⁻² at this potential. As shown in Fig. R2, during 12 hours, the Raman spectra show obvious NiO characteristic feature. Even after conducting 25-hour measurement, the obvious NiO characteristic feature can still be observed, further evidencing that the NiO will not disappear.

We added more discussion on pages 11-12 in the revised manuscript and added the Fig. R2 on pages 20 in the revised Supplementary Information.

Fig. R2 (Supplementary Fig. 13 in the revised Supplementary Information) | *Operando* Raman spectra of NiS electrode in CA measurement for 25 hours at a constant applied potential of -0.3 V vs. RHE.

Q1.3. *Third, the main findings and discussion in the work were based on the transformation of Ni₃S₂/NiO interfaces, including the formation process and the enhanced HER mechanism of the interfaces. However, at a large HER current density, the content of NiO was gradually decreased and possibly disappeared for a long-term operation. Therefore, the enhanced mechanism may be different from that proposed in this work when the HER operated for a long-term period (such as the increased content of the S vacancies).*

Response: Thanks a lot for Reviewer #5's critical question. As shown in Figure R2, the long-time *operando* Raman investigations confirm the existence of the NiO and Ni₃S₂ under HER conditions. On this point, the NiO may be responsible for the improvement of catalytic performance. The enhancement of HER performances by appearance of NiO has also been reported (Gong et al., *Nat. Commun.* 2014, **5**, 4695; Subbaraman et al., *Science* 2011, **334**, 1256; Huang et al., *ACS Energy Lett.* 2019, **4**, 3002). The theoretical calculations in our study further elucidate the positive influence of NiO on water dissociation and Ni₃S₂ on facilitating H* coupling, thereby addressing the enhanced HER activity at the interfaces of Ni₃S₂/NiO. Therefore, we believe the NiO phase play an important role for the enhanced HER activity.

Q1.4. *Fourth, "Additionally, the catalytic activity of the NiS-40 sample is very stable, showing no overpotential increase over 25 hours of continuous operation at a current density of 200 mA cm⁻² in 1 M KOH (Supplementary Fig. 5c)." Based on the comment mentioned above, the structure of the HER-post catalysts is suggested to be characterized.*

Response: We have conducted more post characterization of the electrode after conducting 25-hour chronopotentiometric measurement at a current density of 200 mA cm⁻². As shown in Fig. R3a and R3b, the XRD patterns of HER-post catalyst confirm that the pristine NiS transforms into Ni₃S₂, while the Ni 2p_{3/2} XPS spectra show obvious NiO feature. The Ni *K-edge* XAS spectra further show the NiO characteristic feature located at 8350 eV (Fig. R3c), and it can be

fitted by using NiO and Ni₃S₂ reference well (Fig. R3d). The above characterizations strongly evidence that the HER-post catalysts exist in form of Ni₃S₂/NiO. In addition, the spherical aberration corrected transmission electron microscope (SAC-TEM) was also employed to confirm the existence of Ni₃S₂/NiO, which will be discussed in the following response.

We have added the above discussion and please see it on Pages 7 in the revised manuscript and added the Fig R3 on Pages 11 in the revised Supplementary Information.

Fig. R3 (Supplementary Fig. 7 in the revised Supplementary Information) | The characterization of the HER-post catalyst after conducting 25-hour chronopotentiometric measurement at a current density of 200 mA cm⁻²: (a) XRD pattern; (b) Ni 2p_{3/2} XPS spectra; (c) Ni *K-edge* XAS spectra; (d) the Ni *K-edge* XAS of HER-post catalyst consists of a linear combination of spectra of Ni₃S₂ and NiO reference.

Q2. Compared to the most reported catalyst, the HER activity of the Ni₃S₂/NiO interfaces (10 mA cm⁻²@95 mV) in this work had no obvious advantages. In fact, the overpotentials of most reported catalysts in the alkaline conditions were lower than that of Pt/C.

Response: Thanks a lot for Reviewer #5's comment. We agree that the HER activity in our work may have no obvious advantage in comparison with the state-of-the-art catalyst. Nevertheless, the main contribution of our work lies in employing *operando* XAS and Raman spectroscopy to reveal the dynamic phase transition of NiS, a phenomenon not previously identified in the literature. Furthermore, the combination of NAP-XPS and theoretical investigations further unveiled that the Ni₃S₂/NiO hetero-interface would function as the actual active sites to promote water dissociation, and the strong electronic interaction in the interface provide optimized energetics for coupling of H* to form H₂. Therefore, the most valuable aspect of our work is the study of the mechanism through which transition metal chalcogenides

reach high HER catalytic performance. Our work inspired that the chemistry of transition metal chalcogenides is highly dynamic, and a careful control of the working conditions may lead to the *in-situ* formation of catalytic species that boost their catalytic performance. In addition, our experimental methodology, combining *in-situ* spectroscopies, experimental mechanistic studies and DFT calculations, which may be applied for the study of many other challenging metal chalcogenide systems.

To further highlight the importance of our work, we have changed our manuscript title to 'Dynamic restructuring of nickel sulfides for electrocatalytic hydrogen evolution reaction'.

Q3. Based on the HRTEM image (Fig. 1d), the identification of NiO region was lack of evidence.

Response: Thanks a lot for Reviewer #5's comments. The spherical aberration corrected transmission electron microscope (SAC-TEM) with higher resolution was used to characterize the microstructure of the NiS-40 sample after conducting 25-hour chronopotentiometric measurement at a current density of 200 mA cm^{-2} . As shown in Fig. R4, the large-area TEM image reveals numerous small NiO clusters anchored onto Ni₃S₂. The HR-TEM image exhibits a distinct interface that partitions the image into two regions. By analyzing the FFT images (inset) corresponding to these two regions, the cluster region has an interplanar spacing of 0.21 nm corresponding well to the (200) lattice plane of NiO, while the other region has an interplanar spacing of 0.30 nm which matches well with (-110) lattice plane of Ni₃S₂. More importantly, elemental mapping shows the preferential enrichment of oxygen elements on the small cluster, which strongly reveals the small cluster is assigned to be NiO. The SAC-TEM observation results are consistent with the results by the XRD, XPS, *operando* XAS and *operando* Raman investigations.

We have added the discussion and please see it on Pages 7-8 in the revised manuscript and updated the Fig. 1d.

Fig. R4 (Fig. 1d in the revised manuscript) | The SAC-TEM image, high resolution TEM image (inset are the FFT of the both region) and the corresponding elemental mapping of the NiS-40 sample after conducting 25-hour chronopotentiometric measurement at a current density of 200 mA cm^{-2} .

Reviewer #5 (Remarks to the Author):

The revised version of the manuscript was improved addressing all questions. I recommend its acceptance for publication.